# ALIGN-SAM: SEEKING FLATTER MINIMA FOR BETTER CROSS-SUBSET ALIGNMENT

**Van-Anh Nguyen**
Department of Data Science and AI
Monash University, Australia
van-anh.nguyen@monash.edu

**Trung Le**
Department of Data Science and AI
Monash University, Australia
trunglm@monash.edu

**Mehrtash Harandi**
Department of Electrical and Computer Systems Engineering
Monash University, Australia
mehrtash.harandi@monash.edu

**Thanh-Toan Do**
Department of Data Science and AI
Monash University, Australia
toan.do@monash.edu

**Linh Ngo Van**
Computer Science
Hanoi University of Science and Technology, Vietnam
linhnv@soict.hust.edu.vn

**Dinh Phung**
Department of Data Science and AI
Monash University, Australia
dinh.phung@monash.edu

## ABSTRACT

Sharpness-Aware Minimization (SAM) has proven effective in enhancing deep neural network performance by simultaneously minimizing the training loss and the sharpness of the loss landscape, thereby guiding models toward flatter minima that are empirically linked to improved generalization. From another perspective, generalization can be seen as a model's ability to remain stable under distributional variability. In particular, effective learning requires that updates derived from different subsets or resamplings of the same data distribution remain consistent. In this work, we investigate the connection between the flatness induced by SAM and the alignment of gradients across random subsets of the data distribution, and propose *Align-SAM* as a novel strategy to further enhance model generalization. Align-SAM extends the core principles of SAM by promoting optimization toward flatter minima on a primary subset (the training set), while simultaneously enforcing low loss on an auxiliary subset that is drawn from the same distribution. This dual-objective approach leads to solutions that are not only resilient to local perturbations but also robust against distributional shifts in each training iteration. Empirical evaluations demonstrate that Align-SAM consistently improves generalization across diverse datasets and challenging settings, including scenarios with noisy labels and limited data availability.

## 1 INTRODUCTION

Deep neural networks have emerged as the dominant approach for solving complex tasks such as classification, often outperforming traditional machine learning models. These models learn by adjusting a vast number of parameters to minimize prediction errors or maximize task-specific performance. In practice, training is conducted on a finite dataset $\mathcal{S}$, sampled from an unknown underlying distribution $\mathcal{D}$. The quality and alignment of this dataset with the target distribution significantly impact model efficiency and performance Hestness et al. (2017); Kaplan et al. (2020). Despite their ability to learn complex patterns, deep learning models can also capture noise or random fluctuations in training data, leading to overfitting Arpit et al. (2017); Zhang et al. (2016); Liu et al. (2020). This results in excellent performance on training data but poor predictions on new, unseen data, especially with domain shifts. Generalization McAllester (1999); Dziugaite & Roy (2017b), measured by comparing prediction errors on $S$ and $\mathcal{D}$, becomes crucial. Balancing a model's ability to fit training data with its risk of overfitting is a key challenge in machine learning.

Several studies have been done on this problem, both from theoretical and practical perspectives. Statistical learning theory has proposed different complexity measures that are capable of controlling generalization errors (Vapnik, 1998; Bartlett & Mendelson, 2003; Mukherjee et al., 2002; Bousquet & Elisseeff, 2002; Poggio et al., 2004). In general, they develop a bound for the general error on $\mathcal{D}$. Theory suggests that minimizing the intractable general error on $\mathcal{D}$ is equivalent to minimizing the empirical loss on $S$ with some constraints to the complexity of models and training size (Alquier et al., 2016b). An alternative strategy for mitigating generalization errors involves the utilization of an optimizer to learn optimal parameters for models with a specific local geometry. This approach enables models to find wider local minima (*i.e.*, flat minima), which makes them more robust against data shift between training and testing sets (Jiang et al., 2020; Petzka et al., 2021; Huang et al., 2025).

The connection between a model's generalization and the width of minima has been investigated theoretically and empirically in many studies, notably (Hochreiter & Schmidhuber, 1994; Neyshabur et al., 2017; Dinh et al., 2017; Fort & Ganguli, 2019). A specific method within this paradigm is Sharpness-Aware Minimisation (SAM) (Foret et al., 2021), which has emerged as an effective technique for enhancing the generalization ability of deep learning models. SAM seeks a perturbed model within the vicinity of a current model that maximizes the loss over a training set. Eventually, SAM leads the model to the region where both the current model and its perturbation model have low loss values, ensuring flatness. The success of SAM and its variants (Kwon et al., 2021; Kim et al., 2022) has inspired further investigation into its formulation and behavior, as evidenced by recent works such as (Kaddour et al., 2022; Möllenhoff & Khan, 2022; Andriushchenko & Flammarion, 2022a; Ji et al., 2024).

Additionally, inspired by the PAC-Bayes theorem Alquier (2023), SAM provides an upper bound on the generalization loss over the data distribution $\mathcal{D}$ by considering the loss of perturbed models trained on a random dataset $S \sim \mathcal{D}$. This guides optimization toward minimizing the worst-case loss within a neighborhood of parameters. Such a framework has been shown to encourage convergence to flatter minima on the random dataset $S$, which in turn promotes improving generalization. In practice, however, directly minimizing sharpness over the entire training set $S$ would require forward and backward passes on all of $S$, which is computationally infeasible with large datasets. Instead, standard stochastic optimizers, such as SGD or Adam, are applied that rely on randomly sampled mini-batches, thereby reducing sharpness on each subset and stochastically approximating minimization of sharpness over the full dataset $S$.

From a complementary perspective, generalization can be viewed as the model's ability to remain reliable on new subsets drawn from the same data distribution $\mathcal{D}$. Specifically, we adopt the viewpoint that a model exhibits strong generalization if, although optimized primarily on one random subset $S$, it can also perform well on another independently drawn auxiliary subset $S^a$, where $S, S^a \sim \mathcal{D}$. Motivated by this, we formulate our objective as finding models that minimize sharpness and loss on the primary subset while simultaneously maintaining low loss on the auxiliary subset, thereby ensuring stability across resamplings of the data distribution. To achieve this, we propose *Align-SAM*, a novel method that updates model parameters toward solutions that are both flat and low-loss on the primary subset, while maintaining robust performance across auxiliary subsets, thus implicitly promoting stronger generalization to the full distribution.

In summary, our contributions in this work are as follows:

- We approach generalization from a novel perspective by framing it as an alignment across random subsets drawn from the same data distribution. Building on this viewpoint, we propose Align-SAM, a method designed to enhance both model flatness and stability under distributional variability. Align-SAM primarily updates model parameters by guiding them toward regions in parameter space that minimize sharpness and loss on a primary subset, while simultaneously encouraging strong performance on an auxiliary subset sampled independently from the same distribution. This is achieved by leveraging a combination of gradients computed on both subsets during optimization.

- We demonstrate the effectiveness of Align-SAM in enhancing generalization performance across a variety of settings. Our evaluation begins with image classification tasks, covering both training from scratch and transfer learning on datasets ranging from small to large scale. We further assess its robustness under noisy label conditions with varying noise levels. Additionally, we extend our experiments to meta-learning scenarios to evaluate Align-SAM's ability to generalize beyond meta-training tasks and adapt across diverse

domains. The consistent performance gains across these experiments show that Align-SAM not only improves robustness to label noise and generalization across tasks but also promotes more stable and reliable predictions in varied settings.

## 2 RELATED WORKS

**Sharpness-Aware Minimization.** The correlation between the wider minima and the generalization capacity has been extensively explored both theoretically and empirically in various studies Tan et al. (2025); Jiang et al. (2020); Petzka et al. (2021); Dziugaite & Roy (2017a); Zhuang et al. (2022); Kwon et al. (2021). Many works suggested that finding flat minimizers might help to reduce generalization error and increase robustness to data distributional shift problems in various settings Jiang et al. (2020); Petzka et al. (2021); Huang et al. (2025). There are multiple works have explored the impact of different training parameters, including batch size, learning rate, gradient covariance, and dropout, on the flatness of discovered minima such as Keskar et al. (2017); Jastrzebski et al. (2017); Wei et al. (2020); Deng et al. (2025).

Sharpness-aware minimization (SAM) (Foret et al., 2021) is a recent optimization technique designed to improve the generalization error of neural networks by considering the sharpness of the loss landscape during training. SAM minimizes the worst-case loss around the current model and effectively updates models towards flatter minima to achieve low training loss and maximize generalization performance on new and unseen data. SAM has been successfully applied to various tasks and domains, such as vision models (Chen et al., 2021), language models (Bahri et al., 2022), federated learning (Qu et al., 2022; Xing et al., 2025), Bayesian Neural Networks (Nguyen et al., 2023), domain generalization (Cha et al., 2021), multi-task learning (Phan et al., 2022) and meta-learning bi-level optimization (Abbas et al., 2022). Multiple varieties of SAM have been developed to address limitations of the original method, including ASAM (Kwon et al., 2021), Friendly-SAM (Li et al., 2024), GSAM (Du et al., 2022), VASSO (Li & Giannakis, 2024), and other curvature- or alignment-aware extensions. Efficiency-oriented approaches such as SAF Du et al. (2022) approximate SAM's perturbation to reduce computational overhead, GNAM Zhang et al. (2023) promotes first-order flatness through gradient-norm regularization. LookSAM Liu et al. (2022) further improves efficiency by reusing the same perturbation direction across $k$ consecutive iterations, significantly lowering SAM's computation while maintaining its flatness-seeking behaviour. Recent work has also been inspired by the Lookahead optimizer to deploy multi-step strategies to explore flatter regions of the loss landscape: Lookahead-SAM (Yu et al., 2024) integrates Lookahead's extrapolation–interpolation mechanism with SAM to reach wider minima, whereas Lookbehind-SAM (Mordido et al., 2024) incorporates backward steps before the SAM update to better navigate sharp regions.

**Implicit Biases and Behaviors of SAM.** SAM was inspired by the PAC-Bayes theorem Alquier et al. (2016a); Alquier (2023); Alquier et al. (2016b), which provides an upper bound on generalization loss and motivates the pursuit of flat minima. Several works have since sought to better understand and improve Sharpness-Aware Minimization (SAM). Andriushchenko and Flammarion Andriushchenko & Flammarion (2022b) offer theoretical insights into SAM's optimization dynamics, emphasizing its implicit regularization effects. Compagnoni et al. Monzio Compagnoni et al. (2023) model SAM using a stochastic differential equation (SDE), providing a continuous-time interpretation of its behavior. Wen et al. (Wen et al., 2023b) study how SAM reduces sharpness during optimization, while Chen et al. Chen et al. (2023) show that SAM generalizes better than SGD by avoiding sharp minima. To further improve SAM, Luo et al. (Luo et al., 2024) introduce explicit eigenvalue regularization to control curvature during training. Finally, Wen et al. (Wen et al., 2023a) argue that the generalization benefits of sharpness-aware algorithms stem not only from sharpness minimization but also from other favorable inductive biases.

A complementary strand of work examines data distribution, class imbalance, and long-tailed regimes, where sharpness varies significantly across classes. Nguyen et al. (2024) analyze the features learned under SAM and show that SAM encourages models to rely on both simple and complex features. They further demonstrate that modifying the training data distribution to reduce simplicity bias improves in-distribution generalization. In class-imbalanced settings, several methods adapt SAM to better handle head–tail disparities. ImbSAM Zhou et al. (2023a) incorporates imbalance-aware perturbations to prevent minority classes from being overshadowed by head-class gradients. Class-Conditional SAM Zhou et al. (2023b) aligns SAM's perturbation with class-specific curvature to reduce sharpness for

tail classes, improving long-tailed accuracy. Focal-SAM Li et al. (2025) integrates focal reweighting with SAM to emphasize hard or misclassified instances while maintaining stable optimization.

# 3 PROPOSED METHOD

**Notions.** We start by introducing the notions used throughout our paper. We denote $\mathcal{D}$ as the data/label distribution to generate pairs of data/label $(x, y)$. Given a model with the model parameter $\theta$, we denote the per-sample loss induced by $(x, y)$ as $\ell(x, y; \theta)$. Let $S$ be a random subset drawn from the distribution $\mathcal{D}$. We denote the *empirical* and *generalization* losses as $\mathcal{L}_S(\theta) = \mathbb{E}_S[\ell(x, y; \theta)]$ and $\mathcal{L}_\mathcal{D}(\theta) = \mathbb{E}_\mathcal{D}[\ell(x, y; \theta)]$ respectively. We define $\mathcal{L}_\mathcal{D}(\theta \mid S)$ as an *upper bound defined over $S$* of the general loss $\mathcal{L}_\mathcal{D}(\theta)$. Note that inspired by SAM (Foret et al., 2021), we use the sharpness over $S$ to define $\mathcal{L}_\mathcal{D}(\theta \mid S)$ (see Theorem 1). Finally, we use $|A|$ to denote the cardinality of a set $A$.

## 3.1 PROBLEM FORMULATION

Given a random subset $S^t$ whose examples are sampled from $\mathcal{D}$ (*i.e.*, $S^t \sim \mathcal{D}^{N_t}$ with $N_t = |S^t|$), we use $\mathcal{L}_\mathcal{D}(\theta \mid S^t)$ to train models. $S^t$ is known as the training set. Among the models that minimize this loss, we select the one that minimizes the general loss as follows:

$$\min_{\theta^*} \mathcal{L}_\mathcal{D}(\theta^*) \text{ s.t. } \theta^* \in \mathcal{A}_\mathcal{D}(S^t) := \text{argmin}_\theta \mathcal{L}_\mathcal{D}(\theta \mid S^t). \tag{1}$$

We note that $\mathcal{A}_\mathcal{D}(S^t)$ returns the optimal models $\theta^*$ that minimizes the upper bound $\mathcal{L}_\mathcal{D}(\theta \mid S^t)$. Among the set of such minimizers $\theta^*$, we select the one that further minimizes the true generalization loss $\mathcal{L}_\mathcal{D}$. The reason for the formulation in (1) is that although $\mathcal{L}_\mathcal{D}(\theta \mid S^t)$ is an upper bound of the general loss $\mathcal{L}_\mathcal{D}(\theta)$, there always exists a gap between them. Therefore, the additional outer minimization helps to refine the solutions.

We now denote $S^a$ (*i.e.*, $S^a \sim \mathcal{D}^{N_a}$ with $N_a = |S^a|$) as an other random subset sampled from $\mathcal{D}$, $S^a$ is called the auxiliary set. With respect to this auxiliary set, we have the following theorem.

**Theorem 1.** *Under conditions $\mathcal{L}_D(\theta) \leq E_{\epsilon_i \sim N(0,\rho)} \mathcal{L}_D(\theta + \epsilon)$ similar to SAM (Foret et al., 2021), with a probability greater than $1 - \delta$ (i.e., $\delta \in [0, 1]$) over the choice of $S^a \sim \mathcal{D}^{N_a}$, we then have for any optimal models $\theta^* \in \mathcal{A}_\mathcal{D}(S^t)$:*

$$\mathcal{L}_\mathcal{D}(\theta^*) \leq \mathcal{L}_\mathcal{D}(\theta^* \mid S^a) + \frac{8L}{\sqrt{N_a}} \sqrt{\log\left(\frac{N_a + k}{\delta}\right)} + \frac{4L}{\sqrt{N_a}} \mathcal{O}(1)$$

$$+ \frac{4L}{\sqrt{N_a}} k \log\left(1 + \frac{\|\theta^*\|^2}{\rho}\left(1 + \sqrt{\log\left(\frac{N_a}{k}\right)}\right)\right). \tag{2}$$

*where we denote $\mathcal{L}_\mathcal{D}(\theta^* \mid S) := \max_{\theta' : \|\theta' - \theta^*\|_2 \leq \rho} \mathcal{L}_S(\theta')$ for any random subset $S \sim \mathcal{D}^N$ (i.e $S^t, S^a$), and $L$ is the upper bound of the loss function (i.e., $\ell(x, y; \theta) \leq L, \forall x, y, \theta$), $k$ is the model size as the length of vector $\theta$, and $\rho > 0$ is the perturbation radius.*

According to Theorem 1, $\mathcal{L}_\mathcal{D}(\theta^* \mid S) := \max_{\theta' : \|\theta' - \theta^*\|_2 \leq \rho} \mathcal{L}_S(\theta')$ can be viewed as an upper bound of the generalization loss $\mathcal{L}_\mathcal{D}(\theta^*)$, up to a constant difference. Moreover, our theorem 1 (see Appendix A.1 for proof) can be viewed as an extension of Theorem 1 in Foret et al. (2021), where we apply the PAC-Bayes theorem from Alquier et al. (2016a) to prove an upper bound for the generalization loss of any bounded loss, instead of the 0-1 loss in Foret et al. (2021). We can generalize this proof for $S^t$ to explain why we use $\mathcal{L}_\mathcal{D}(\theta \mid S^t) := \max_{\theta' : \|\theta' - \theta\|_2 \leq \rho} \mathcal{L}_{S^t}(\theta')$ as an objective to minimize, as in (1).

Based on Theorem 1, we can rewrite the objectives in (1) as:

$$\min_{\theta^*} \mathcal{L}_\mathcal{D}(\theta^* \mid S^a) \text{ s.t. } \theta^* \in \mathcal{A}_\mathcal{D}(S^t) := \text{argmin}_\theta \mathcal{L}_\mathcal{D}(\theta \mid S^t), \tag{3}$$

where $\mathcal{L}_\mathcal{D}(\theta \mid S) := \max_{\theta' : \|\theta' - \theta\|_2 \leq \rho} \mathcal{L}_S(\theta')$. Among all models that minimize the upper generalization bound on a random training subset $S^t$, we select the one that further minimizes the upper

generalization bound on an independently drawn auxiliary subset $S^a$. In other words, the optimal solution is the one that achieves low sharpness and loss on the primary subset while simultaneously maintaining low loss on the auxiliary subset.

Our theory works for $S^t, S^a \sim \mathcal{D}$, where $\mathcal{D}$ is the distribution to generate $(x, y)$. In the practical version of Algorithm 1, we replace $\mathcal{D}$ by the empirical distribution $S$ and at each iteration, we sample two mini-batches $B^t, B^a \sim S$. Because when the training size (i.e. $|S|$) approaches $\infty$, the distribution $S$ asymptotically converges to the distribution $\mathcal{D}$. Using stochastic optimization, we reformulate this objective into an iterative update scheme, where the model is trained with two independently drawn mini-batches, $B^t$ and $B^a$, such that each update encourages convergence toward flat minima while aligning performance across subsets in every iteration.

## 3.2 OUR SOLUTION

Our motivation here is to primarily optimize the loss over the training set $S^t$, while using $S^a$ to further enhance the generalization ability, where both $S^t$ and $S^a$ are random subsets drawn from the same data distribution. Our formulation in (3) has the same form as a bi-level optimization problem similar to MAML (Finn et al., 2017), developed for meta-learning. Inspired by MAML, a naive approach would be to consider $f(\theta) := \arg\min_\theta \mathcal{L}_\mathcal{D}(\theta \mid S^t)$ (*i.e.*, $f(\theta) = \theta - \eta \nabla_\theta \mathcal{L}_\mathcal{D}(\theta \mid S^t)$) and finding $\theta^* := \arg\min_\theta \mathcal{L}_\mathcal{D}(f(\theta) \mid S^a)$ with respect to $\theta$. However, this naive approach *does not align* with our objective, as it mainly focuses on optimizing the loss $\mathcal{L}_\mathcal{D}(f(\theta) \mid S^a)$ over the auxiliary set $S^a$, in which, the auxiliary set acts like the validation set in MAML. Here we note that in Franceschi et al. (2018), bi-level optimization was employed to learn optimal hyperparameters (e.g., the weight of a regularizer) by finding hyperparameters such that a model trained on a training set performs well on a validation set. This is fundamentally different from our aim, which is to study how to achieve flat minima that align two independent random subsets in every update step. Moreover, directly adapting the MAML bi-level formulation would require distinct training and validation sets, which are often unavailable in most scenarios. For these reasons, both our theoretical framework and technical approach differ substantially from those in Franceschi et al. (2018).

Using stochastic optimization, we reformulate the objective (3) into an iterative update scheme, where the model is trained using two independently drawn mini-batches $B^t$ and $B^a$, with both batches sampled independently from the training set $S$. The *Align-SAM* is presented as follows: *at each iteration, our primary objective is to optimize $\mathcal{L}(\theta \mid B^t)$, primarily based on its gradients, in such a way that future models are able to implicitly perform well on $B^a$.* To achieve this, similar to SAM (Foret et al., 2021), we approximate $\mathcal{L}(\theta \mid B^t) = \max_{\|\theta' - \theta\| \le \rho} \mathcal{L}_{B^t}(\theta') \approx \mathcal{L}_{B^t}(\theta + \eta_1 \nabla \mathcal{L}_{B^t}(\theta))$ for a sufficient small learning rate $\eta_1 > 0$ (i.e., $\eta_1 \| \nabla \mathcal{L}_{B^t}(\theta) \| \le \rho$) and $\mathcal{L}(\theta \mid B^a) = \max_{\|\theta' - \theta\| \le \rho} \mathcal{L}_{B^a}(\theta') \approx \mathcal{L}_{B^a}(\theta + \eta_2 \nabla \mathcal{L}_{B^a}(\theta))$ for a sufficient small learning rate $\eta_2 > 0$ (i.e., $\eta_2 \| \nabla \mathcal{L}_{B^a}(\theta) \| \le \rho$).

At each iteration, while primarily using the gradients of $\mathcal{L}(\theta \mid B^t)$ for optimization, we also utilize the gradient of $\mathcal{L}(\theta \mid B^a)$ to ensure congruent behavior between these two gradients. Specifically, at the $l$-th iteration, we update as follows:

$$\tilde{\theta}_l^a = \theta_l + \eta_2 \nabla_\theta \mathcal{L}_{B^a}(\theta_l), \tag{4}$$

$$\tilde{\theta}_l^t = \theta_l + \eta_1 \nabla_\theta \mathcal{L}_{B^t}(\theta_l) - \eta_2 \nabla_\theta \mathcal{L}_{B^a}\left(\tilde{\theta}_l^a\right), \tag{5}$$

$$\theta_{l+1} = \theta_l - \eta \nabla_\theta \mathcal{L}_{B^t}\left(\tilde{\theta}_l^t\right), \tag{6}$$

where $\eta_1 > 0, \eta_2 > 0$, and $\eta > 0$ are the learning rates, while $\mathcal{L}_{B^t}(\theta_l)$ and $\mathcal{L}_{B^a}(\theta_l)$ represent the empirical losses over the mini-batches $B^t, B^a \sim S^t$ respectively.

According to (6) (*i.e.*, $\theta_{l+1} = \theta_l - \eta \nabla_\theta \mathcal{L}_{B^t}\left(\tilde{\theta}_l^t\right)$), $\theta_{l+1}$ is updated to minimize $\mathcal{L}_{B^t}\left(\tilde{\theta}_l^t\right)$. We now do first-order Taylor expansion for $\mathcal{L}_{B^t}\left(\tilde{\theta}_l^t\right)$ as

$$\mathcal{L}_{B^t}\left(\tilde{\theta}_l^t\right) \approx \mathcal{L}_{B^t}(\theta_l) + \eta_1 \|\nabla_\theta \mathcal{L}_{B^t}(\theta_l)\|_2^2 - \eta_2 \nabla_\theta \mathcal{L}_{B^t}(\theta_l) \cdot \nabla_\theta \mathcal{L}_{B^a}\left(\tilde{\theta}_l^a\right), \tag{7}$$

where $\cdot$ specifies the dot product.

From (7), we reach the conclusion that the update in (6) aims to *minimize* simultaneously **(i)** $\mathcal{L}_{B^t}(\theta_l)$, **(ii)** $\|\nabla_\theta \mathcal{L}_{B^t}(\theta_l)\|_2^2$, and *maximize* **(iii)** $\nabla_\theta \mathcal{L}_{B^t}(\theta_l) \cdot \nabla_\theta \mathcal{L}_{B^a}\left(\tilde{\theta}_l^a\right)$. While the effects in (i) and (ii) are similar to SAM (Foret et al., 2021), maximizing $\nabla_\theta \mathcal{L}_{B^t}(\theta_l) \cdot \nabla_\theta \mathcal{L}_{B^a}\left(\tilde{\theta}_l^a\right))$ encourages two gradients of the losses over $B^t$ and $B^a$ to become more congruent. The following theorem shows that, during training, the two gradients of interest become *more congruent*.

**Theorem 2.** *For sufficiently small learning rates* $\eta_1 \leq \dfrac{\left|\nabla_\theta \mathcal{L}_{B^t}(\theta_l) \cdot \nabla_\theta \mathcal{L}_{B^a}\left(\tilde{\theta}_l^a\right)\right|}{12 \left|\nabla_\theta \mathcal{L}_{B^a}\left(\tilde{\theta}_l^a\right)^T H_{B^t}(\theta_l) \nabla_\theta \mathcal{L}_{B^t}(\theta_l)\right|}$ *and* $\eta_2 \leq$

$\min \left\{ \dfrac{\left|\nabla_\theta \mathcal{L}_{B^t}(\theta_l) \cdot \nabla_\theta \mathcal{L}_{B^a}\left(\tilde{\theta}_l^a\right)\right|}{6 \left|\nabla_\theta \mathcal{L}_{B^a}\left(\tilde{\theta}_l^a\right)^T H_{B^t}(\theta_l) \nabla_\theta \mathcal{L}_{B^a}\left(\tilde{\theta}_l^a\right)\right|}, \dfrac{\left|\nabla_\theta \mathcal{L}_{B^t}(\theta_l) \cdot \nabla_\theta \mathcal{L}_{B^a}\left(\tilde{\theta}_l^a\right)\right|}{6 \left|\nabla_\theta \mathcal{L}_{B^a}\left(\tilde{\theta}_l^a\right)^T H_{B^a}\left(\tilde{\theta}_l^a\right) \nabla_\theta \mathcal{L}_{B^t}(\theta_l)\right|} \right\}$, *we have*

$$
\nabla_\theta \mathcal{L}_{B^t}\left(\tilde{\theta}_l^t\right) \cdot \nabla_\theta \mathcal{L}_{B^a}\left(\tilde{\theta}_l^a\right) \geq \begin{cases} \frac{1}{2} \nabla_\theta \mathcal{L}_{B^t}(\theta_l) \cdot \nabla_\theta \mathcal{L}_{B^a}\left(\tilde{\theta}_l^a\right) & if \nabla_\theta \mathcal{L}_{B^t}(\theta_l) \cdot \nabla_\theta \mathcal{L}_{B^a}\left(\tilde{\theta}_l^a\right) \geq 0 \\ \frac{3}{2} \nabla_\theta \mathcal{L}_{B^t}(\theta_l) \cdot \nabla_\theta \mathcal{L}_{B^a}\left(\tilde{\theta}_l^a\right) & otherwise \end{cases}
$$

$$(8)$$

Theorem 2 (see Appendix A.1 for proof) indicates that two gradients $\nabla_\theta \mathcal{L}_{B^t}\left(\tilde{\theta}_l^t\right)$ and $\nabla_\theta \mathcal{L}_{B^a}\left(\tilde{\theta}_l^a\right)$ are encouraged to be more congruent since our update aims to maximize its lower bound $c \times \nabla_\theta \mathcal{L}_{B^t}(\theta_l) \cdot \nabla_\theta \mathcal{L}_{B^a}\left(\tilde{\theta}_l^a\right)$ (i.e., $c = 0.5$ or $c = 1.5$).

**Practical Algorithm.** Inspired by SAM (Foret et al., 2021), we set $\eta_2 = \frac{\rho}{\|\nabla_\theta \mathcal{L}_{B^a}(\theta_l)\|_2}$ and $\eta_1 = \lambda \frac{\rho}{\|\nabla_\theta \mathcal{L}_{B^t}(\theta_l)\|_2}$, where $\rho > 0$ are perturbation radius and $\lambda$ is trade-off coefficient for combining gradient from $B^t$ and $B^a$. In practice, we observe that setting $\lambda > 1$, which prioritizes the gradient from the training mini-batch $B^t$, results in improved performance. This trade-off is discussed in Section A.3.

The pseudo-code of Align-SAM is presented in Algorithm 1. Compared to standard SAM, our method requires additional forward and backward passes due to the use of an auxiliary batch. To reduce computational overhead, we substitute the gradient of purturbation model on the auxiliary set $\nabla_\theta \mathcal{L}_{B^a}\left(\tilde{\theta}_l^a\right)$ with the gradient of the current model $\nabla_\theta \mathcal{L}_{B^a}(\theta_l)$ in Equation 5. This reuse of the gradient maintains the primary objective of AlignSAM: maximizing loss on the training set while simultaneously minimizing it on the auxiliary set. Additionally, we set the auxiliary batch size $|B^a|$ significantly smaller than the primary training batch size $|B^t|$, ensuring that most computation is devoted to the main update step. As a result, Align-SAM is only marginally slower than standard SAM, as reported in Table 9. Further details are provided in the Appendix.

## 3.3 Convergence Analysis

It is well known that the normalized (practical) version of SAM *does not converge* to the minimizer of the training loss, as rigorously demonstrated in Si & Yun (2023) (Theorem 4.6), one of the most comprehensive analyses of SAM's convergence behavior. Our proposed approach shares the *same convergence rate* as standard SAM, as established in Si & Yun (2023) (Theorem 4.6). Details in Appendix A.2

## 4 Experiments

In this section, we present the results of various experiments to evaluate the effectiveness of our Align-SAM, including training from scratch, transfer learning on different dataset sizes, learning with noisy labels, and a meta-learning setting.

## 4.1 Image Classification From Scratch

We first conduct experiments on ImageNet, Food101, and CIFAR datasets with standard image classification settings trained from scratch. The performance is compared with baseline models trained

---

**Algorithm 1** Pseudo-code of Align-SAM

---

1: **Input:** $\rho, \lambda, \eta$, the number of iterations $T$, and the training set $S$
2: **Output:** the optimal model $\theta_T$.
3: **for** $l = 1$ to $T$ **do**
4:     Sample mini-batches $B^t, B^a \sim S$.
5:     $g_a = \nabla_\theta \mathcal{L}_{B^a}(\theta_l)$
6:     $g_t = \nabla_\theta \mathcal{L}_{B^t}(\theta_l)$
7:     Compute $\tilde{\theta}_l^t \leftarrow \theta_l + \rho \left( \lambda \frac{g_t}{\|g_t\|_2} - \frac{g_a}{\|g_a\|_2} \right)$.
8:     Compute $\theta_{l+1} \leftarrow \theta_l - \eta \nabla_\theta \mathcal{L}_{B^t}\left( \tilde{\theta}_l^t \right)$.
9: **end for**

---

with the SGD, SAM, ASAM, and the integration of ASAM and Align-SAM. For all experiments of Align-SAM, we consistently set $\lambda = 2$ and discuss the effect of this trade-off in Section A.3.

**ImageNet dataset.** We use ResNet18 and ResNet34 models for experiments on the ImageNet dataset, with an input size of $224 \times 224$. For all experiments with Align-SAM, we consistently set $\lambda = 2$, while the perturbation radius $\rho$ is configured according to the SAM method. Specifically, in this experiment, we set $\rho = 0.1$ for both SAM and Align-SAM. The models are trained for 200 epochs with basic data augmentations (random cropping, horizontal flipping, and normalization). We use an initial learning rate of 0.1, a batch size of 2048 for the training mini-batches, and 512 for the auxiliary mini-batches, following a cosine learning schedule across all experiments in this paper. We extend this experiment to the mid-sized Food101 dataset using the same settings, except for a batch size of 128 for the training and 32 for the auxiliary mini-batches. Performance results are detailed in Table 1.

Table 1: Classification accuracy on the ImageNet and Food101 datasets. All models are trained from scratch with 200 epochs.

| Dataset | Method | Resnet18 | | Resnet34 | |
|---|---|---|---|---|---|
| | | Top-1 | Top-5 | Top-1 | Top-5 |
| ImageNet | SAM | 62.46 | 84.19 | 63.73 | 84.95 |
| | Align-SAM | **63.64** | **85.22** | **65.89** | **86.84** |
| Food101 | SAM | 73.15 | 89.85 | 73.87 | 90.84 |
| | Align-SAM | **73.45** | **90.35** | **74.47** | **91.27** |

**CIFAR dataset.** We used three architectures: WideResnet28x10, Pyramid101, and Densenet121 with an input size of $32 \times 32$ for CIFAR datasets. To replicate the baseline experiments, we followed the hyperparameters provided in the original papers. Specifically, for CIFAR-100, we set $\rho = 0.1$, and for CIFAR-10, we used $\rho = 0.05$ for SAM, VASSO and Align-SAM. The same procedure and settings were applied to ASAM and Align-ASAM, with the perturbation radius $\rho = 1.0$ for CIFAR-100 and $\rho = 0.5$ for CIFAR-10. Other training configurations are consistent with those used in the ImageNet experiments, except for data augmentations (horizontal flipping, four-pixel padding, and random cropping). We use $\theta = 0.9$ as the default parameter for VASSO. The results are reproduced and reported in Tables 2, while the SGD results are referenced from Foret et al. (2021).

Our proposed method outperforms the baselines across various settings. On both ImageNet and Food101, it significantly surpasses the baselines, with a notable improvement in both Top-1 and Top-5 accuracy. For CIFAR-10, performance is close to the saturation point, making further improvements challenging. Nevertheless, Align-SAM achieves slight enhancements across all cases. On CIFAR-100, where models are more prone to overfitting compared to CIFAR-10, Align-SAM still delivers competitive results.

Table 2: Classification accuracy on the CIFAR datasets. All models are trained from scratch three times with different random seeds and we report the mean and standard deviation of accuracies.

| Method | WideResnet28x10 | Pyramid101 | Densenet121 |
|---|---|---|---|
| **Dataset CIFAR-100** | | | |
| SGD Foret et al. (2021) | $81.20 \pm 0.200$ | $80.30 \pm 0.300$ | - |
| SAM Foret et al. (2021) | $83.00 \pm 0.035$ | $81.99 \pm 0.636$ | $68.72 \pm 0.409$ |
| VASSO Li & Giannakis (2024) | $83.11 \pm 0.063$ | $82.04 \pm 0.127$ | $69.00 \pm 0.261$ |
| GSAM Zhuang et al. (2022) | $83.13 \pm 0.099$ | $81.87 \pm 0.143$ | $68.88 \pm 0.201$ |
| LookSAM (k=1) Liu et al. (2022) | $82.89 \pm 0.111$ | $82.25 \pm 0.273$ | $69.05 \pm 0.182$ |
| **Align-SAM** | $\mathbf{83.72 \pm 0.049}$ | $\mathbf{82.53 \pm 0.282}$ | $\mathbf{69.10 \pm 0.311}$ |
| ASAM | $83.16 \pm 0.296$ | $82.02 \pm 0.134$ | $69.62 \pm 0.120$ |
| Align-ASAM | $\mathbf{83.88 \pm 0.042}$ | $\mathbf{82.31 \pm 0.183}$ | $\mathbf{69.71 \pm 0.339}$ |
| **Dataset CIFAR-10** | | | |
| SGD Foret et al. (2021) | $96.50 \pm 0.100$ | $96.00 \pm 0.100$ | - |
| SAM Foret et al. (2021) | $96.87 \pm 0.027$ | $96.17 \pm 0.174$ | $91.28 \pm 0.241$ |
| VASSO Li & Giannakis (2024) | $96.84 \pm 0.014$ | $96.22 \pm 0.035$ | $91.18 \pm 0.063$ |
| GSAM Zhuang et al. (2022) | $96.91 \pm 0.020$ | $96.15 \pm 0.113$ | $91.50 \pm 0.109$ |
| LookSAM(k=1) Liu et al. (2022) | $\mathbf{97.00 \pm 0.181}$ | $96.00 \pm 0.207$ | $91.10 \pm 0.196$ |
| **Align-SAM** | $96.91 \pm 0.007$ | $\mathbf{96.47 \pm 0.219}$ | $\mathbf{91.54 \pm 0.307}$ |
| ASAM Kwon et al. (2021) | $96.91 \pm 0.063$ | $96.45 \pm 0.042$ | $\mathbf{92.04 \pm 0.240}$ |
| **Align-ASAM** | $\mathbf{97.15 \pm 0.063}$ | $\mathbf{96.56 \pm 0.261}$ | $92.02 \pm 0.000$ |

## 4.2 TRANSFER LEARNING

In this subsection, we further evaluate Align-SAM in the transfer learning setting using the ImageNet pre-trained models to fine-tune both small-size, mid-size, and large-size datasets. All initialized weights are available on the Pytorch library.

Table 3: Transfer learning on ImageNet with Resnet models.

| Model | Top-1 Acc | | Top-5 Acc | |
|---|---|---|---|---|
| | SAM Foret et al. (2021) | Align-SAM | SAM Foret et al. (2021) | Align-SAM |
| Resnet18 | $70.44 \pm 0.12$ | $\mathbf{70.92 \pm 0.05}$ | $89.63 \pm 0.04$ | $\mathbf{89.90 \pm 0.04}$ |
| Resnet34 | $73.06 \pm 0.48$ | $\mathbf{73.94 \pm 0.14}$ | $91.29 \pm 0.03$ | $\mathbf{91.81 \pm 0.03}$ |
| Resnet50 | $75.17 \pm 0.23$ | $\mathbf{75.91 \pm 0.19}$ | $92.58 \pm 0.01$ | $\mathbf{92.83 \pm 0.01}$ |
| ViT-Adapter-S | $79.96 \pm 0.30$ | $\mathbf{80.04 \pm 0.32}$ | $95.35 \pm 0.01$ | $\mathbf{95.39 \pm 0.03}$ |

First, we conduct experiments on ImageNet using three models from the ResNet family and a ViT-Adapter-S (which incorporates lightweight Adapter modules with a plain ViT-Small backbone). The ResNet models are pre-trained on ImageNet, while the backbone ViT-Small of ViT-Adapter-S is pre-trained on ImageNet-21k. Each model is then fine-tuned for 50 epochs using either SAM or Align-SAM with a learning rate of $0.01$. We set $\rho = 0.05$ for SAM and Align-SAM; basic augmentation techniques are the same as training from the scratch setting. Results reported in Table 3 show that our methods outperform baselines with a significant gap in both top-1 and top-5 accuracies.

Next, we examine this setting on small and mid-sized datasets on three models of the EfficientNet family. We fine-tune with a learning rate of $0.05$ in 50 epochs and use $\rho = 0.1$ for all experiments of SAM, VASSO (with $\theta = 0.9$ as the default) Li & Giannakis (2024), and Align-SAM. In Table 4, Align-SAM achieves a noticeable improvement compared to most of the baselines on all small-size, mid-size, and large-size datasets, demonstrating its robustness and stability across various experiment settings.

## 4.3 TRAIN WITH NOISY LABEL

In addition to mitigating data shifts between training and testing datasets, we evaluate the robustness of Align-SAM against noisy labels on the standard training procedure. Specifically, we adopt a classical noisy-label setting for CIFAR-10 and CIFAR-100, in which a portion of the training set's

Table 4: Transfer learning accuracy of small and medium datasets. All models are fine-tuned from pre-trained weights on ImageNet.

| Dataset | Top-1 Acc | | | | Top-5 Acc | | | |
|---|---|---|---|---|---|---|---|---|
| | SGD | SAM | VASSO | Align-SAM | SGD | SAM | VASSO | Align-SAM |
| **EfficientNet-B2** | | | | | | | | |
| Stanford Cars | $89.14 \pm 0.11$ | $89.68 \pm 0.17$ | $89.91 \pm 0.24$ | $\mathbf{90.39 \pm 0.07}$ | $97.60 \pm 0.20$ | $98.04 \pm 0.07$ | $98.03 \pm 0.09$ | $\mathbf{98.30 \pm 0.09}$ |
| FGVC-Aircraft | $85.83 \pm 0.23$ | $86.25 \pm 0.36$ | $86.23 \pm 0.22$ | $\mathbf{87.22 \pm 0.27}$ | $95.72 \pm 0.02$ | $95.87 \pm 0.06$ | $95.94 \pm 0.05$ | $\mathbf{95.94 \pm 0.03}$ |
| Oxford IIIT Pets | $92.17 \pm 0.19$ | $92.34 \pm 0.11$ | $92.40 \pm 0.16$ | $\mathbf{92.64 \pm 0.17}$ | $99.23 \pm 0.02$ | $99.35 \pm 0.02$ | $99.34 \pm 0.01$ | $\mathbf{99.35 \pm 0.07}$ |
| Flower102 | $95.06 \pm 0.01$ | $95.22 \pm 0.14$ | $95.37 \pm 0.11$ | $\mathbf{95.43 \pm 0.10}$ | $99.08 \pm 0.18$ | $99.11 \pm 0.19$ | $\mathbf{99.23 \pm 0.09}$ | $99.18 \pm 0.02$ |
| Food101 | $83.50 \pm 0.01$ | $85.12 \pm 0.07$ | $84.65 \pm 0.03$ | $\mathbf{85.51 \pm 0.02}$ | $96.10 \pm 0.32$ | $96.83 \pm 0.08$ | $96.60 \pm 0.09$ | $\mathbf{97.14 \pm 0.00}$ |
| Country211 | $11.94 \pm 0.14$ | $12.48 \pm 0.03$ | $12.49 \pm 0.10$ | $\mathbf{13.41 \pm 0.00}$ | $23.70 \pm 0.13$ | $25.49 \pm 0.07$ | $24.90 \pm 0.13$ | $\mathbf{27.06 \pm 0.16}$ |
| **EfficientNet-B3** | | | | | | | | |
| Stanford Cars | $89.01 \pm 0.19$ | $89.40 \pm 0.09$ | $89.55 \pm 0.12$ | $\mathbf{89.86 \pm 0.14}$ | $97.73 \pm 0.21$ | $98.03 \pm 0.07$ | $98.01 \pm 0.05$ | $\mathbf{98.10 \pm 0.01}$ |
| FGVC-Aircraft | $84.88 \pm 0.08$ | $85.19 \pm 0.11$ | $85.15 \pm 0.19$ | $\mathbf{85.78 \pm 0.25}$ | $95.53 \pm 0.12$ | $95.67 \pm 0.00$ | $95.67 \pm 0.02$ | $\mathbf{96.26 \pm 0.10}$ |
| Oxford IIIT Pets | $92.68 \pm 0.25$ | $92.58 \pm 0.02$ | $92.64 \pm 0.06$ | $\mathbf{92.75 \pm 0.19}$ | $99.00 \pm 0.01$ | $99.19 \pm 0.05$ | $99.23 \pm 0.10$ | $\mathbf{99.26 \pm 0.11}$ |
| Flower102 | $94.59 \pm 0.10$ | $94.73 \pm 0.14$ | $94.94 \pm 0.17$ | $\mathbf{95.32 \pm 0.26}$ | $98.95 \pm 0.08$ | $99.12 \pm 0.16$ | $99.13 \pm 0.07$ | $\mathbf{99.26 \pm 0.07}$ |
| Food101 | $83.75 \pm 0.12$ | $85.79 \pm 0.13$ | $85.69 \pm 0.14$ | $\mathbf{85.95 \pm 0.13}$ | $96.22 \pm 0.02$ | $97.12 \pm 0.00$ | $97.07 \pm 0.09$ | $\mathbf{97.33 \pm 0.00}$ |
| Country211 | $12.96 \pm 0.01$ | $13.38 \pm 0.09$ | $13.36 \pm 0.08$ | $\mathbf{13.61 \pm 0.05}$ | $26.11 \pm 0.56$ | $25.78 \pm 0.08$ | $25.91 \pm 0.05$ | $\mathbf{26.71 \pm 0.26}$ |
| **EfficientNet-B4** | | | | | | | | |
| Stanford Cars | $84.72 \pm 0.04$ | $85.08 \pm 0.16$ | $85.06 \pm 0.07$ | $\mathbf{85.46 \pm 0.32}$ | $96.41 \pm 0.07$ | $96.45 \pm 0.01$ | $96.53 \pm 0.04$ | $\mathbf{96.81 \pm 0.00}$ |
| FGVC-Aircraft | $79.95 \pm 0.61$ | $79.96 \pm 0.04$ | $80.02 \pm 0.38$ | $\mathbf{80.53 \pm 0.51}$ | $94.87 \pm 0.08$ | $94.65 \pm 0.08$ | $94.68 \pm 0.13$ | $\mathbf{94.74 \pm 0.01}$ |
| Oxford IIIT Pets | $91.89 \pm 0.13$ | $92.02 \pm 0.23$ | $92.04 \pm 0.18$ | $\mathbf{92.07 \pm 0.00}$ | $99.28 \pm 0.10$ | $99.43 \pm 0.07$ | $\mathbf{99.59 \pm 0.08}$ | $99.44 \pm 0.02$ |
| Flower102 | $92.73 \pm 0.04$ | $93.02 \pm 0.14$ | $93.02 \pm 0.16$ | $\mathbf{93.07 \pm 0.16}$ | $98.49 \pm 0.07$ | $98.68 \pm 0.02$ | $\mathbf{98.73 \pm 0.07}$ | $98.63 \pm 0.05$ |
| Food101 | $84.55 \pm 0.14$ | $86.13 \pm 0.06$ | $86.18 \pm 0.10$ | $\mathbf{86.40 \pm 0.44}$ | $96.31 \pm 0.03$ | $97.07 \pm 0.01$ | $97.07 \pm 0.03$ | $\mathbf{97.31 \pm 0.02}$ |
| Country211 | $14.63 \pm 0.09$ | $14.80 \pm 0.13$ | $14.97 \pm 0.11$ | $\mathbf{15.26 \pm 0.16}$ | $27.60 \pm 0.00$ | $28.09 \pm 1.77$ | $28.00 \pm 0.18$ | $\mathbf{28.24 \pm 0.14}$ |

labels are symmetrically flipped with noise fractions {0.2, 0.4, 0.6, 0.8}, while the testing set's labels remain unchanged.

Table 5: Results under label noise on CIFAR datasets with ResNet32. Each experiment is conducted three times using different random seeds, and we report their averages and standard deviations.

| Method | Noise rate (%) | | | |
|---|---|---|---|---|
| | 0.2 | 0.4 | 0.6 | 0.8 |
| **Dataset CIFAR-100** | | | | |
| SGD | $66.22 \pm 0.355$ | $59.26 \pm 0.045$ | $46.77 \pm 0.020$ | $26.49 \pm 0.640$ |
| SAM | $66.16 \pm 0.721$ | $59.95 \pm 0.622$ | $50.81 \pm 0.353$ | $24.26 \pm 1.209$ |
| FSAM | $65.73 \pm 0.219$ | $58.96 \pm 0.381$ | $49.36 \pm 1.103$ | $25.92 \pm 1.173$ |
| VASSO ($\theta = 0.9$) | $66.52 \pm 0.254$ | $59.67 \pm 0.318$ | $50.09 \pm 0.353$ | $20.85 \pm 0.077$ |
| VASSO ($\theta = 0.2$) | $65.16 \pm 0.042$ | $59.07 \pm 0.820$ | $48.35 \pm 1.046$ | $\mathbf{28.49 \pm 0.551}$ |
| Align-SAM | $\mathbf{66.78 \pm 0.657}$ | $\mathbf{60.78 \pm 0.636}$ | $\mathbf{51.03 \pm 0.502}$ | $27.66 \pm 1.265$ |
| ASAM | $66.88 \pm 0.593$ | $61.53 \pm 0.487$ | $52.77 \pm 0.561$ | $30.33 \pm 1.788$ |
| Align-ASAM | $\mathbf{67.38 \pm 0.106}$ | $\mathbf{62.72 \pm 0.304}$ | $\mathbf{54.58 \pm 0.572}$ | $\mathbf{32.77 \pm 0.388}$ |
| **Dataset CIFAR-10** | | | | |
| SGD | $89.98 \pm 0.070$ | $84.83 \pm 0.085$ | $75.06 \pm 0.385$ | $54.47 \pm 1.265$ |
| SAM | $91.26 \pm 0.007$ | $88.19 \pm 1.060$ | $83.43 \pm 0.622$ | $61.69 \pm 0.289$ |
| FSAM | $91.35 \pm 0.318$ | $87.58 \pm 0.353$ | $82.78 \pm 2.057$ | $58.09 \pm 2.276$ |
| VASSO ($\theta = 0.9$) | $91.47 \pm 0.487$ | $88.17 \pm 0.890$ | $83.75 \pm 0.480$ | $67.71 \pm 4.129$ |
| VASSO ($\theta = 0.2$) | $90.45 \pm 0.855$ | $86.28 \pm 0.997$ | $77.33 \pm 0.806$ | $\mathbf{70.95 \pm 0.770}$ |
| Align-SAM | $\mathbf{92.38 \pm 0.007}$ | $\mathbf{90.20 \pm 0.318}$ | $\mathbf{85.33 \pm 0.268}$ | $70.02 \pm 0.403$ |
| ASAM | $91.98 \pm 0.007$ | $89.24 \pm 0.572$ | $84.39 \pm 0.445$ | $64.82 \pm 6.880$ |
| Align-ASAM | $\mathbf{92.06 \pm 0.367}$ | $\mathbf{90.01 \pm 0.282}$ | $\mathbf{86.09 \pm 0.657}$ | $\mathbf{73.25 \pm 0.353}$ |

All experiments are conducted using the ResNet32 architecture, with models trained from scratch for 200 epochs. The batch size is set to 512 for the training mini-batches and 128 for the auxiliary mini-batches. Following Foret et al. (2021), we set $\rho = 0.1$ SAM, FSAM Li et al. (2024), VASSO Li & Giannakis (2024), and Align-SAM, $\rho = 1.0$ for ASAM and Align-ASAM when training with all noise levels. Exceptionally, for the setting with 80% noisy labels, the perturbation radius for SAM, FSAM, Align-SAM, ASAM, and Align-ASAM is reduced by half to ensure more stable convergence. This observation is also noted in Li et al. (2024) and Foret et al. (2021).

In line with Li et al. (2024), we apply additional cutout techniques along with the basic augmentations outlined in Section 4.1. We report the results of VASSO with $\theta = \{0.2, 0.9\}$, as presented in their paper Li & Giannakis (2024), where $\theta = 0.2$ is expected to yield better performance. However, we observe that $\theta = 0.2$ performs better only in the setting with 80% noisy labels, while in other noisy label settings, it gives a lower result compared to $\theta = 0.9$. Each experiment is repeated three times with different random seeds, and we report the average and standard deviation of the results in Table 5. Note that training with SGD is prone to overfitting as the number of epochs increases. Therefore, we present the best results for SGD training at both 200 and 400 epochs.

## 5 ABLATION STUDY

### 5.1 COSINE SIMILARITY OF GRADIENTS

We present the cosine similarity of gradients before and update the model using SAM and Align-SAM in Figure 1. Detail analysis is presented in Appendix A.3.

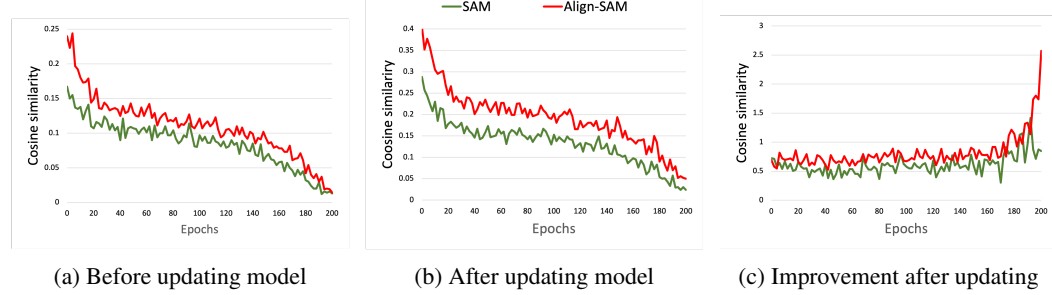

(a) Before updating model    (b) After updating model    (c) Improvement after updating

Figure 1: Cosine similarity of two gradients $\nabla_\theta \mathcal{L}_{B^t}(\theta_l)$ and $\nabla_\theta \mathcal{L}_{B^a}\left(\tilde{\theta}_l^a\right)$ (**a**) before updating model, (**b**) after updating model and (**c**) the improvement of this similarity.

**The size of the auxiliary subset** $|B^a|$ **and complexity, the sensitivity of trade-off coefficient** $\lambda$**, the analysis of loss landscape**, and details of these experiments are presented in Appendix A.3.

## 6 CONCLUSION AND LIMITATION

In conclusion, this work revisits Sharpness-Aware Minimization (SAM) through the lens of cross-subset alignment, offering a fresh perspective on generalization. While SAM encourages flat minima to improve generalization, we argue that effective generalization also hinges on the alignment between two independently sampled subsets from the same data distribution. Building on this insight, we introduce *Align-SAM* to find models that minimize sharpness and loss on the primary subset while simultaneously maintaining low loss on the auxiliary subset, thereby ensuring stability across resamplings of the data distribution. By explicitly aligning the optimization process across both subsets, Align-SAM produces models that are not only robust to perturbations but also more resilient to distributional shifts. Extensive experiments confirm that Align-SAM delivers consistent gains in generalization, particularly under challenging conditions such as label noise and data scarcity. One limitation to note is that using an additional auxiliary subset in each training iteration may increase training time (depending on the size of the auxiliary sets). We view this as a trade-off between performance and training complexity. However, this issue could potentially be mitigated by reusing the gradients from the previous steps. We leave this as a direction for future work to reduce training complexity while maintaining performance.

### ACKNOWLEDGMENT

Trung Le, Mehrtash Harandi, and Dinh Phung were supported by the ARC Discovery Project grant DP250100262. Additionally, Dinh Phung further acknowledged the support from the Australian Research Council (ARC) Discovery Project DP230101176. Trung Le and Mehrtash Harandi were also supported by the Air Force Office of Scientific Research under award number FA9550-23-S-0001. This research/work was supported by Monash eResearch capabilities, including M3.

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

## A APPENDIX / SUPPLEMENTAL MATERIAL

In this appendix, we present the proofs in our paper and additional experiments. We open-source our code and provide instruction, scripts, and log files to reproduce experiments at `https://anonymous.4open.science/r/AlignSAM-43CD/README.md`

### A.1 ALL PROOFS

**Proof of Theorem 1**

*Proof.* We use the PAC-Bayes theory in this proof. In PAC-Bayes theory, $\theta$ could follow a distribution, says $P$, thus we define the expected loss over $\theta$ distributed by $P$ as follows:

$$\mathcal{L}_{\mathcal{D}}(\theta, P) = \mathbb{E}_{\theta \sim P}\big[\mathcal{L}_{\mathcal{D}}(\theta)\big]$$
$$\mathcal{L}_{\mathcal{S}}(\theta, P) = \mathbb{E}_{\theta \sim P}\big[\mathcal{L}_{\mathcal{S}}(\theta)\big].$$

For any distribution $P = \mathcal{N}(\mathbf{0}, \sigma_P^2 \mathbb{I}_k)$ and $Q = \mathcal{N}(\theta, \sigma^2 \mathbb{I}_k)$ over $\theta \in \mathbb{R}^k$, where $P$ is the prior distribution and $Q$ is the posterior distribution, use the PAC-Bayes theorem in Alquier et al. (2016a), for all $\beta > 0$, with a probability at least $1 - \delta$, we have

$$\mathcal{L}_{\mathcal{D}}(\theta, Q) \leq \mathcal{L}_{\mathcal{S}}(\theta, Q) + \frac{1}{\beta}\Big[\mathsf{KL}(Q\|P) + \log\frac{1}{\delta} + \Psi(\beta, N)\Big], \tag{9}$$

where $\Psi$ is defined as

$$\Psi(\beta, N) = \log \mathbb{E}_P \mathbb{E}_{\mathcal{D}^N}\Big[\exp\big\{\beta\big[\mathcal{L}_{\mathcal{D}}(f_\theta) - \mathcal{L}_{\mathcal{S}}(f_\theta)\big]\big\}\Big].$$

When the loss function is bounded by $L$, then

$$\Psi(\beta, N) \leq \frac{\beta^2 L^2}{8N}.$$

The task is to minimize the second term of RHS of (9), we thus choose $\beta = \sqrt{8N}\frac{\mathsf{KL}(Q\|P) + \log\frac{1}{\delta}}{L}$. Then the second term of RHS of (9) is equal to

$$\sqrt{\frac{\mathsf{KL}(Q\|P) + \log\frac{1}{\delta}}{2N}} \times L.$$

The KL divergence between $Q$ and $P$, when they are Gaussian, is given by formula

$$\mathsf{KL}(Q\|P) = \frac{1}{2}\left[\frac{k\sigma^2 + \|\theta\|^2}{\sigma_P^2} - k + k\log\frac{\sigma_P^2}{\sigma^2}\right].$$

For given posterior distribution $Q$ with fixed $\sigma^2$, to minimize the KL term, the $\sigma_P^2$ should be equal to $\sigma^2 + \|\theta\|^2/k$. In this case, the KL term is no less than

$$k\log\Big(1 + \frac{\|\theta_0\|^2}{k\sigma^2}\Big).$$

Thus, the second term of RHS is

$$\sqrt{\frac{\mathsf{KL}(Q\|P) + \log\frac{1}{\delta}}{2N}} \times L \geq \sqrt{\frac{k\log\big(1 + \frac{\|\theta\|^2}{k\sigma^2}\big)}{4N}} \times L \geq L$$

when $\|\theta\|^2 > \sigma^2\big\{\exp(4N/k) - 1\big\}$. Hence, for any $\|\theta\|_2 > \sigma^2\big\{\exp(4N/k) - 1\big\}$, we have the RHS is greater than the LHS, and the inequality is trivial. In this work, we only consider the case:

$$\|\theta\|^2 < \sigma^2\big(\exp\{4N/k\} - 1\big). \tag{10}$$

Distribution $P$ is Gaussian centered around $\mathbf{0}$ with variance $\sigma_P^2 = \sigma^2 + \|\theta\|^2/k$, which is unknown at the time we set up the inequality, since $\theta$ is unknown. Meanwhile, we have to specify $P$ in advance,

since $P$ is the prior distribution. To deal with this problem, we could choose a family of $P$ such that its means cover the space of $\theta$ satisfying inequality (10). We set

$$c = \sigma^2 \big(1 + \exp\{4N/k\}\big)$$

$$P_j = \mathcal{N}\big(0, c \exp \frac{1-j}{k} \mathbb{I}_k\big)$$

$$\mathfrak{P} := \big\{P_j : j = 1, 2, \dots \big\}$$

Then the following inequality holds for a particular distribution $P_j$ with probability $1 - \delta_j$ with $\delta_j = \frac{6\delta}{\pi^2 j^2}$

$$\mathbb{E}_{\theta' \sim \mathcal{N}(\theta, \sigma^2)} \mathcal{L}_{\mathcal{D}}\big(f_{\theta'}\big) \leq \mathbb{E}_{\theta' \sim \mathcal{N}(\theta, \sigma^2)} \mathcal{L}_{\mathcal{S}}\big(f_{\theta'}\big) + \frac{1}{\beta} \bigg[ \mathsf{KL}(Q \| P_j) + \log \frac{1}{\delta_j} + \Psi(\beta, N) \bigg].$$

Use the well-known equation: $\sum_{j=1}^{\infty} \frac{1}{j^2} = \frac{\pi^2}{6}$, then with probability $1 - \delta$, the above inequality holds with every $j$. We pick

$$j^* := \bigg\lfloor 1 - k \log \frac{\sigma^2 + \|\theta\|^2/k}{c} \bigg\rfloor = \bigg\lfloor 1 - k \log \frac{\sigma^2 + \|\theta\|^2/k}{\sigma^2(1 + \exp\{4N/k\})} \bigg\rfloor.$$

Therefore,

$$1 - j^* = \bigg\lceil k \log \frac{\sigma^2 + \|\theta\|^2/k}{c} \bigg\rceil$$

$$\Rightarrow \quad \log \frac{\sigma^2 + \|\theta\|^2/k}{c} \leq \frac{1 - j^*}{k} \leq \log \frac{\sigma^2 + \|\theta_0\|^2/k}{c} + \frac{1}{k}$$

$$\Rightarrow \quad \sigma^2 + \|\theta\|^2/k \leq c \exp\bigg\{\frac{1 - j^*}{k}\bigg\} \leq \exp(1/k)\big[\sigma^2 + \|\theta\|^2/k\big]$$

$$\Rightarrow \quad \sigma^2 + \|\theta\|^2/k \leq \sigma_{P_{j^*}}^2 \leq \exp(1/k)\big[\sigma^2 + \|\theta\|^2/k\big].$$

Thus the KL term could be bounded as follow

$$\mathsf{KL}(Q \| P_{j^*}) = \frac{1}{2} \bigg[ \frac{k\sigma^2 + \|\theta\|^2}{\sigma_{P_{j^*}}^2} - k + k \log \frac{\sigma_{P_{j^*}}^2}{\sigma^2} \bigg]$$

$$\leq \frac{1}{2} \bigg[ \frac{k(\sigma^2 + \|\theta\|^2/k)}{\sigma^2 + \|\theta\|^2/k} - k + k \log \frac{\exp(1/k)(\sigma^2 + \|\theta\|^2/k)}{\sigma^2} \bigg]$$

$$= \frac{1}{2} \bigg[ k \log \frac{\exp(1/k)(\sigma^2 + \|\theta\|^2/k)}{\sigma^2} \bigg]$$

$$= \frac{1}{2} \bigg[ 1 + k \log \bigg(1 + \frac{\|\theta_0\|^2}{k\sigma^2}\bigg) \bigg]$$

For the term $\log \frac{1}{\delta_{j^*}}$, with recall that $c = \sigma^2\big(1 + \exp(4N/k)\big)$ and

$j^* = \bigg\lfloor 1 - k \log \frac{\sigma^2 + \|\theta\|^2/k}{\sigma^2(1 + \exp\{4N/k\})} \bigg\rfloor$, we have

$$\log \frac{1}{\delta_{j^*}} = \log \frac{(j^*)^2 \pi^2}{6\delta} = \log \frac{1}{\delta} + \log \bigg(\frac{\pi^2}{6}\bigg) + 2\log(j^*)$$

$$\leq \log \frac{1}{\delta} + \log \frac{\pi^2}{6} + 2\log \bigg(1 + k \log \frac{\sigma^2(1 + \exp(4N/k))}{\sigma^2 + \|\theta\|^2/k}\bigg)$$

$$\leq \log \frac{1}{\delta} + \log \frac{\pi^2}{6} + 2\log \bigg(1 + k \log \big(1 + \exp(4N/k)\big)\bigg)$$

$$\leq \log \frac{1}{\delta} + \log \frac{\pi^2}{6} + 2\log \bigg(1 + k\big(1 + \frac{4N}{k}\big)\bigg)$$

$$\leq \log \frac{1}{\delta} + \log \frac{\pi^2}{6} + \log(1 + k + 4N).$$

Hence, the inequality

$$\mathcal{L}_{\mathcal{D}}\Big(\theta', \mathcal{N}(\theta, \sigma^2 \mathbb{I}_k)\Big) \le \mathcal{L}_{\mathcal{S}}\Big(\theta', \mathcal{N}(\theta, \sigma^2 \mathbb{I}_k)\Big) + \sqrt{\frac{\mathsf{KL}(Q \| P_{j^*}) + \log \frac{1}{\delta_{j^*}}}{2N}} \times L$$

$$\le \mathcal{L}_{\mathcal{S}}\Big(\theta', \mathcal{N}(\theta, \sigma^2 \mathbb{I}_k)\Big)$$

$$+ \frac{L}{2\sqrt{N}} \sqrt{1 + k \log\Big(1 + \frac{\|\theta\|^2}{k\sigma^2}\Big) + 2 \log \frac{\pi^2}{6\delta} + 4\log(N+k)}$$

$$\le \mathcal{L}_{\mathcal{S}}\Big(\theta', \mathcal{N}(\theta, \sigma^2 \mathbb{I}_k)\Big)$$

$$+ \frac{L}{2\sqrt{N}} \sqrt{k \log\Big(1 + \frac{\|\theta\|^2}{k\sigma^2}\Big) + O(1) + 2 \log \frac{1}{\delta} + 4\log(N+k)}.$$

Since $\|\theta' - \theta\|^2$ is $k$ chi-square distribution, for any positive $t$, we have

$$\mathbb{P}\big(\|\theta' - \theta\|^2 - k\sigma^2 \ge 2\sigma^2\sqrt{kt} + 2t\sigma^2)\big) \le \exp(-t).$$

By choosing $t = \frac{1}{2}\log(N)$, with probability $1 - N^{-1/2}$, we have

$$\|\theta' - \theta\|^2 \le \sigma^2 \log(N) + k\sigma^2 + \sigma^2\sqrt{2k\log(N)} \le k\sigma^2\Big(1 + \sqrt{\frac{\log(N)}{k}}\Big)^2.$$

By setting $\sigma = \rho \times \big(\sqrt{k} + \sqrt{\log(N)}\big)^{-1}$, we have $\|\theta' - \theta\|^2 \le \rho^2$. Hence, we get

$$\mathcal{L}_{\mathcal{S}}\Big(\theta', \mathcal{N}(\theta, \sigma^2 \mathbb{I}_k)\Big) = \mathbb{E}_{\theta \sim \mathcal{N}(\theta, \sigma^2 \mathbb{I}_k)} \mathbb{E}_{\mathcal{S}}\big[f_{\theta'}\big] = \int_{\|\theta' - \theta\| \le \rho} \mathbb{E}_{\mathcal{S}}\big[f_{\theta'}\big] d\mathcal{N}(\theta, \sigma^2 \mathbb{I})$$

$$+ \int_{\|\theta' - \theta\| > \rho} \mathbb{E}_{\mathcal{S}}\big[f_{\theta'}\big] d\mathcal{N}(\theta, \sigma^2 \mathbb{I})$$

$$\le \Big(1 - \frac{1}{\sqrt{N}}\Big) \max_{\|\theta' - \theta\| \le \rho} \mathcal{L}_{\mathcal{S}}(\theta') + \frac{1}{\sqrt{N}} L$$

$$\le \max_{\|\theta' - \theta\|_2 \le \rho} \mathcal{L}_{\mathcal{S}}(\theta') + \frac{2L}{\sqrt{N}}.$$

It follows that

$$\mathcal{L}_{\mathcal{D}}(\theta) \le \max_{\|\theta' - \theta\| \le \rho} \mathcal{L}_{\mathcal{S}}(\theta') + \frac{4L}{\sqrt{N}} \Bigg[ \sqrt{k \log\Big(1 + \frac{\|\theta\|^2}{\rho^2}\big(1 + \sqrt{\log(N)/k}\big)^2\Big)}$$

$$+ 2\sqrt{\log\big(\frac{N+k}{\delta}\big) + O(1)} \Bigg]$$

$$= \mathcal{L}_{\mathcal{D}}(\theta \mid \mathcal{S}) + \frac{4L}{\sqrt{N}} \Bigg[ \sqrt{k \log\Big(1 + \frac{\|\theta\|^2}{\rho^2}\big(1 + \sqrt{\log(N)/k}\big)^2\Big)}$$

$$+ 2\sqrt{\log\big(\frac{N+k}{\delta}\big) + O(1)} \Bigg].$$

By choosing $\theta = \theta^*$, which is the solution on a random subset $S^t$ and $\mathcal{S} = S^a$, which is another subset, $S^t, S^a \sim \mathcal{D}$, hence $N = N^a$, we reach the conclusion. $\qquad\square$

**Proof of Theorem 2**

*Proof.* We have

$$\mathcal{L}_{B^t}\Big(\tilde{\theta}_l^t\Big) = \mathcal{L}_{B_t}(\theta_l) + \eta_1 \|\nabla_\theta \mathcal{L}_{B^t}(\theta_l)\|_2^2 - \eta_2 \nabla_\theta \mathcal{L}_{B^t}(\theta_l) \cdot \nabla_\theta \mathcal{L}_{B^a}\Big(\tilde{\theta}_l^a\Big).$$

This follows that

$$\nabla_\theta \mathcal{L}_{B^t}\left(\tilde{\theta}_l^t\right) = \nabla_\theta \mathcal{L}_{B_t}\left(\theta_l\right) + 2\eta_1 H_{B^t}\left(\theta_l\right)\nabla_\theta \mathcal{L}_{B^t}\left(\theta_l\right)$$
$$- \eta_2\left[H_{B^t}\left(\theta_l\right)\nabla_\theta \mathcal{L}_{B^a}\left(\tilde{\theta}_l^a\right) + H_{B^a}\left(\tilde{\theta}_l^a\right)\nabla_\theta \mathcal{L}_{B^t}\left(\theta_l\right)\right],$$

where $H_{B^t}\left(\theta_l\right) = \nabla_\theta^2 \mathcal{L}_{B_t}\left(\theta_l\right)$ and $H_{B^a}\left(\tilde{\theta}_l^a\right) = \nabla_\theta^2 \mathcal{L}_{B^a}\left(\tilde{\theta}_l^a\right)$ are the Hessian matrices.

$$\nabla_\theta \mathcal{L}_{B^a}\left(\tilde{\theta}_l^a\right) \cdot \nabla_\theta \mathcal{L}_{B^t}\left(\tilde{\theta}_l^t\right) = \nabla_\theta \mathcal{L}_{B_t}\left(\theta_l\right) \cdot \nabla_\theta \mathcal{L}_{B^a}\left(\tilde{\theta}_l^a\right)$$
$$+ 2\eta_1 \nabla_\theta \mathcal{L}_{B^a}\left(\tilde{\theta}_l^a\right)^T H_{B^t}\left(\theta_l\right)\nabla_\theta \mathcal{L}_{B^t}\left(\theta_l\right)$$
$$- \eta_2 \nabla_\theta \mathcal{L}_{B^a}\left(\tilde{\theta}_l^a\right)^T H_{B^t}\left(\theta_l\right)\nabla_\theta \mathcal{L}_{B^a}\left(\tilde{\theta}_l^a\right)$$
$$- \eta_2 \nabla_\theta \mathcal{L}_{B^a}\left(\tilde{\theta}_l^a\right)^T H_{B^a}\left(\tilde{\theta}_l^a\right)\nabla_\theta \mathcal{L}_{B^t}\left(\theta_l\right).$$

We now choose $\eta_1 \leq \dfrac{\left|\nabla_\theta \mathcal{L}_{B_t}(\theta_l)\cdot\nabla_\theta \mathcal{L}_{B^a}\left(\tilde{\theta}_l^a\right)\right|}{12\left|\nabla_\theta \mathcal{L}_{B^a}\left(\tilde{\theta}_l^a\right)^T H_{B^t}(\theta_l)\nabla_\theta \mathcal{L}_{B^t}(\theta_l)\right|}$, we then have

$$\eta_1 \left|\nabla_\theta \mathcal{L}_{B^a}\left(\tilde{\theta}_l^a\right)^T H_{B^t}\left(\theta_l\right)\nabla_\theta \mathcal{L}_{B^t}\left(\theta_l\right)\right| \leq \frac{1}{12}\left|\nabla_\theta \mathcal{L}_{B_t}\left(\theta_l\right)\cdot\nabla_\theta \mathcal{L}_{B^a}\left(\tilde{\theta}_l^a\right)\right|.$$

This further implies

$$\eta_1 \nabla_\theta \mathcal{L}_{B^a}\left(\tilde{\theta}_l^a\right)^T H_{B^t}\left(\theta_l\right)\nabla_\theta \mathcal{L}_{B^t}\left(\theta_l\right) \geq -\frac{1}{12}\left|\nabla_\theta \mathcal{L}_{B_t}\left(\theta_l\right)\cdot\nabla_\theta \mathcal{L}_{B^a}\left(\tilde{\theta}_l^a\right)\right|.$$

Next we choose $\eta_2 \leq \min\left\{\dfrac{\left|\nabla_\theta \mathcal{L}_{B_t}(\theta_l)\cdot\nabla_\theta \mathcal{L}_{B^a}\left(\tilde{\theta}_l^a\right)\right|}{6\left|\nabla_\theta \mathcal{L}_{B^a}\left(\tilde{\theta}_l^a\right)^T H_{B^t}(\theta_l)\nabla_\theta \mathcal{L}_{B^a}\left(\tilde{\theta}_l^a\right)\right|}, \dfrac{\left|\nabla_\theta \mathcal{L}_{B_t}(\theta_l)\cdot\nabla_\theta \mathcal{L}_{B^a}\left(\tilde{\theta}_l^a\right)\right|}{6\left|\nabla_\theta \mathcal{L}_{B^a}\left(\tilde{\theta}_l^a\right)^T H_{B^a}\left(\tilde{\theta}_l^a\right)\nabla_\theta \mathcal{L}_{B^t}(\theta_l)\right|}\right\}$,
we then have

$$\eta_2 \left|\nabla_\theta \mathcal{L}_{B^a}\left(\tilde{\theta}_l^a\right)^T H_{B^t}\left(\theta_l\right)\nabla_\theta \mathcal{L}_{B^a}\left(\tilde{\theta}_l^a\right)\right| \leq \frac{\left|\nabla_\theta \mathcal{L}_{B_t}\left(\theta_l\right)\cdot\nabla_\theta \mathcal{L}_{B^a}\left(\tilde{\theta}_l^a\right)\right|}{6}.$$

$$-\eta_2 \nabla_\theta \mathcal{L}_{B^a}\left(\tilde{\theta}_l^a\right)^T H_{B^t}\left(\theta_l\right)\nabla_\theta \mathcal{L}_{B^a}\left(\tilde{\theta}_l^a\right) \geq -\frac{\left|\nabla_\theta \mathcal{L}_{B_t}\left(\theta_l\right)\cdot\nabla_\theta \mathcal{L}_{B^a}\left(\tilde{\theta}_l^a\right)\right|}{6}.$$

$$\eta_2 \left|\nabla_\theta \mathcal{L}_{B^a}\left(\tilde{\theta}_l^a\right)^T H_{B^a}\left(\tilde{\theta}_l^a\right)\nabla_\theta \mathcal{L}_{B^t}\left(\theta_l\right)\right| \leq \frac{\left|\nabla_\theta \mathcal{L}_{B_t}\left(\theta_l\right)\cdot\nabla_\theta \mathcal{L}_{B^a}\left(\tilde{\theta}_l^a\right)\right|}{6}.$$

$$-\eta_2 \nabla_\theta \mathcal{L}_{B^a}\left(\tilde{\theta}_l^a\right)^T H_{B^a}\left(\tilde{\theta}_l^a\right)\nabla_\theta \mathcal{L}_{B^t}\left(\theta_l\right) \geq -\frac{\left|\nabla_\theta \mathcal{L}_{B_t}\left(\theta_l\right)\cdot\nabla_\theta \mathcal{L}_{B^a}\left(\tilde{\theta}_l^a\right)\right|}{6}.$$

Finally, we yield

$$\nabla_\theta \mathcal{L}_{B^a}\left(\tilde{\theta}_l^a\right) \cdot \nabla_\theta \mathcal{L}_{B^t}\left(\tilde{\theta}_l^t\right) \geq \nabla_\theta \mathcal{L}_{B_t}\left(\theta_l\right)\cdot\nabla_\theta \mathcal{L}_{B^a}\left(\tilde{\theta}_l^a\right) - \frac{1}{2}\left|\nabla_\theta \mathcal{L}_{B_t}\left(\theta_l\right)\cdot\nabla_\theta \mathcal{L}_{B^a}\left(\tilde{\theta}_l^a\right)\right|.$$

$\square$

### Proof of Theorem 3

*Proof.* We first denote $\hat{\theta}_l^t = \theta_l + (\rho_1 - \rho_2)\dfrac{\nabla \mathcal{L}_S(\theta_l)}{\|\nabla \mathcal{L}_S(\theta_l)\|}$. Using the $\beta$-smoothness, we have

$$\mathcal{L}_S\left(\theta_{l+1}\right) \leq \mathcal{L}_S\left(\theta_l\right) + \nabla \mathcal{L}_S\left(\theta_l\right)\cdot\left(\theta_{l+1} - \theta_l\right) + \frac{\beta}{2}\|\theta_{l+1} - \theta_l\|^2$$
$$\leq \mathcal{L}_S\left(\theta_l\right) - \eta\nabla \mathcal{L}_S\left(\theta_l\right)\cdot\nabla \mathcal{L}_{B^t}\left(\tilde{\theta}_l^t\right) + \frac{\beta\eta^2}{2}\|\nabla \mathcal{L}_{B^t}\left(\tilde{\theta}_l^t\right)\|^2.$$

Taking the expectation, we gain

$$
\begin{aligned}
\mathbb{E}\left[\mathcal{L}_S\left(\theta_{l+1}\right)\right] \leq & \mathbb{E}\left[\mathcal{L}_S\left(\theta_l\right)\right] - \eta\mathbb{E}\left[\nabla\mathcal{L}_S\left(\theta_l\right)\cdot\nabla\mathcal{L}_{B^t}\left(\tilde{\theta}_l^t\right)\right] + \frac{\beta\eta^2}{2}\mathbb{E}\left[\|\nabla\mathcal{L}_{B^t}\left(\tilde{\theta}_l^t\right)\|^2\right] \\
\leq & \mathbb{E}\left[\mathcal{L}_S\left(\theta_l\right)\right] - \eta\mathbb{E}\left[\|\nabla\mathcal{L}_S\left(\theta_l\right)\|^2\right] - \eta\mathbb{E}\left[\nabla\mathcal{L}_S\left(\theta_l\right)\cdot\left[\nabla\mathcal{L}_{B^t}\left(\tilde{\theta}_l^t\right) - \nabla\mathcal{L}_S\left(\theta_l\right)\right]\right] \\
& + \frac{\beta\eta^2}{2}\mathbb{E}\left[\|\nabla\mathcal{L}_{B^t}\left(\tilde{\theta}_l^t\right)\|^2\mid\right] \\
= & \mathbb{E}\left[\mathcal{L}_S\left(\theta_l\right)\right] - \eta\mathbb{E}\left[\|\nabla\mathcal{L}_S\left(\theta_l\right)\|^2\right] - \eta\mathbb{E}\left[\nabla\mathcal{L}_S\left(\theta_l\right)\cdot\left[\nabla\mathcal{L}_{B^t}\left(\tilde{\theta}_l^t\right) - \nabla\mathcal{L}_{B^t}\left(\hat{\theta}_l^t\right)\right]\right] \\
& + \frac{\beta\eta^2}{2}\mathbb{E}\left[\|\nabla\mathcal{L}_{B^t}\left(\tilde{\theta}_l^t\right)\|^2\mid\right] - \eta\mathbb{E}\left[\nabla\mathcal{L}_S\left(\theta_l\right)\cdot\left[\nabla\mathcal{L}_S\left(\hat{\theta}_l^t\right) - \nabla\mathcal{L}_S\left(\theta_l\right)\right]\right] \\
\leq & \mathbb{E}\left[\mathcal{L}_S\left(\theta_l\right)\right] - \eta\mathbb{E}\left[\|\nabla\mathcal{L}_S\left(\theta_l\right)\|^2\right] \\
& + \frac{\eta}{2}\mathbb{E}\left[\|\nabla\mathcal{L}_S\left(\theta_l\right)\|^2\right] + \frac{\eta}{2}\mathbb{E}\left[\|\nabla\mathcal{L}_{B^t}\left(\tilde{\theta}_l^t\right) - \nabla\mathcal{L}_{B^t}\left(\hat{\theta}_l^t\right)\|^2\right] \\
& - \eta\mathbb{E}\left[\|\nabla\mathcal{L}_S\left(\theta_l\right)\|\frac{\hat{\theta}_l^t - \theta_l}{\rho_1 - \rho_2}\cdot\left[\nabla\mathcal{L}_S\left(\hat{\theta}_l^t\right) - \nabla\mathcal{L}_S\left(\theta_l\right)\right]\right] + \frac{\beta\eta^2}{2}\mathbb{E}\left[\|\nabla\mathcal{L}_{B^t}\left(\tilde{\theta}_l^t\right)\|^2\right] \\
\leq & \mathbb{E}\left[\mathcal{L}_S\left(\theta_l\right)\right] - \frac{\eta}{2}\mathbb{E}\left[\|\nabla\mathcal{L}_S\left(\theta_l\right)\|^2\right] + \frac{\beta^2\eta}{2}\mathbb{E}\left[\|\tilde{\theta}_l^t - \hat{\theta}_l^t\|^2\right] + \frac{\beta\eta^2}{2}\mathbb{E}\left[\|\nabla\mathcal{L}_{B^t}\left(\tilde{\theta}_l^t\right)\|^2\right] \\
& + \beta\eta\mathbb{E}\left[\frac{\|\nabla\mathcal{L}_S\left(\theta_l\right)\|}{\rho_1 - \rho_2}\|\hat{\theta}_l^t - \theta_l\|^2\right] \\
\leq & \mathbb{E}\left[\mathcal{L}_S\left(\theta_l\right)\right] - \frac{\eta}{2}\mathbb{E}\left[\|\nabla\mathcal{L}_S\left(\theta_l\right)\|^2\right] \\
& + \frac{\beta^2\eta}{2}\mathbb{E}\left[\|\rho_1\frac{\nabla\mathcal{L}_{B^t}\left(\theta_l\right)}{\|\nabla\mathcal{L}_{B^t}\left(\theta_l\right)\|} - \rho_2\frac{\nabla\mathcal{L}_{B^a}\left(\tilde{\theta}_l^a\right)}{\|\nabla\mathcal{L}_{B^a}\left(\tilde{\theta}_l^a\right)\|} - (\rho_1 - \rho_2)\frac{\nabla\mathcal{L}_S\left(\theta_l\right)}{\|\nabla\mathcal{L}_S\left(\theta_l\right)\|}\|^2\right] \\
& + \frac{\beta\eta^2}{2}\mathbb{E}\left[\|\nabla\mathcal{L}_{B^t}\left(\tilde{\theta}_l^t\right)\|^2\right] + \beta\eta\mathbb{E}\left[\frac{\|\nabla\mathcal{L}_S\left(\theta_l\right)\|}{\rho_1 - \rho_2}\|\hat{\theta}_l^t - \theta_l\|^2\right] \\
\leq & \mathbb{E}\left[\mathcal{L}_S\left(\theta_l\right)\right] - \frac{\eta}{2}\mathbb{E}\left[\|\nabla\mathcal{L}_S\left(\theta_l\right)\|^2\right] + 3\beta^2\eta\left(\rho_1^2 + \rho_2^2 - \rho_1\rho_2\right) \\
& + (\rho_1 - \rho_2)\beta\eta\mathbb{E}\left[\|\nabla\mathcal{L}_S\left(\theta_l\right)\|\right] + \frac{\beta\eta^2}{2}\mathbb{E}\left[\|\nabla\mathcal{L}_{B^t}\left(\tilde{\theta}_l^t\right)\|^2\right] \\
\leq & \mathbb{E}\left[\mathcal{L}_S\left(\theta_l\right)\right] - \frac{\eta}{2}\mathbb{E}\left[\|\nabla\mathcal{L}_S\left(\theta_l\right)\|^2\right] + 3\beta^2\eta\left(\rho_1^2 + \rho_2^2 - \rho_1\rho_2\right) \\
& + (\rho_1 - \rho_2)\beta\eta\mathbb{E}\left[\|\nabla\mathcal{L}_S\left(\theta_l\right)\|\right] \\
& + \beta\eta^2\mathbb{E}\left[\|\nabla\mathcal{L}_{B^t}\left(\tilde{\theta}_l^t\right) - \nabla\mathcal{L}_S\left(\tilde{\theta}_l^t\right)\|^2\right] + \beta\eta^2\mathbb{E}\left[\|\nabla\mathcal{L}_S\left(\tilde{\theta}_l^t\right)\|^2\right] \\
\leq & \mathbb{E}\left[\mathcal{L}_S\left(\theta_l\right)\right] - \frac{\eta}{2}\mathbb{E}\left[\|\nabla\mathcal{L}_S\left(\theta_l\right)\|^2\right] + 3\beta^2\eta\left(\rho_1^2 + \rho_2^2 - \rho_1\rho_2\right) \\
& + (\rho_1 - \rho_2)\beta\eta\mathbb{E}\left[\|\nabla\mathcal{L}_S\left(\theta_l\right)\|\right] + \beta\eta^2\left(\sigma^2 + G^2\right) \\
\leq & \mathbb{E}\left[\mathcal{L}_S\left(\theta_l\right)\right] - \frac{\eta}{2}\left(\mathbb{E}\left[\|\nabla\mathcal{L}_S\left(\theta_l\right)\|\right] - \Delta\rho\beta\right)^2 + \frac{1}{2}\eta\Delta\rho^2\beta^2 \\
& + 3\beta^2\eta\left(\rho_1^2 + \rho_2^2 - \rho_1\rho_2\right) + \beta\eta^2\left(\sigma^2 + G^2\right)
\end{aligned}
$$

Rearrange the terms, we obtain

$$
\begin{aligned}
\left(\mathbb{E}\left[\|\nabla\mathcal{L}_S\left(\theta_l\right)\|\right] - \Delta\rho\beta\right)^2 \leq & \frac{2}{\eta}\left(\mathbb{E}\left[\mathcal{L}_S\left(\theta_l\right)\right] - \mathbb{E}\left[\mathcal{L}_S\left(\theta_{l+1}\right)\right]\right) \\
& + \frac{1}{2}\eta\Delta\rho^2\beta^2 + 3\beta^2\eta\left(\rho_1^2 + \rho_2^2 - \rho_1\rho_2\right) + \beta\eta^2\left(\sigma^2 + G^2\right)
\end{aligned}
$$

Take sum $l$ from 1 to $T$, we reach

$$
\begin{aligned}
\frac{1}{T}\sum_{l=1}^{T}\left(\mathbb{E}\left[\|\nabla\mathcal{L}_S\left(\theta_l\right)\|\right]-\Delta\rho\beta\right)^2 \leq & \frac{2}{\eta T}\left(\mathbb{E}\left[\mathcal{L}_S\left(\theta_0\right)\right]-\mathbb{E}\left[\mathcal{L}_S\left(\theta_{T+1}\right)\right]\right)+\Delta\rho^2\beta^2 \\
& +6\beta^2\left(\rho_1^2+\rho_2^2-\rho_1\rho_2\right)+2\beta\eta\left(\sigma^2+G^2\right) \\
\leq & \frac{2}{\eta T}\left(\mathbb{E}\left[\mathcal{L}_S\left(\theta_0\right)\right]-L^*\right)+\Delta\rho^2\beta^2+6\beta^2\left(7\rho_1^2+7\rho_2^2-8\rho_1\rho_2\right) \\
& +2\beta\eta\left(\sigma^2+G^2\right) \\
\leq & \frac{2\Delta}{\eta T}+6\beta^2\left(7\rho_1^2+7\rho_2^2-8\rho_1\rho_2\right)+2\beta\eta\left(\sigma^2+G^2\right)
\end{aligned}
$$

Substitute $\eta=\frac{\sqrt{\Delta}}{\sqrt{\beta T(\sigma^2+G^2)}}$, we arrive at

$$
\frac{1}{T}\sum_{l=1}^{T}\left(\mathbb{E}\left[\|\nabla\mathcal{L}_S\left(\theta_l\right)\|\right]-\Delta\rho\beta\right)^2 \leq \frac{4\sqrt{\beta\Delta\left(\sigma^2+G^2\right)}}{\sqrt{T}}+6\beta^2\left(7\rho_1^2+7\rho_2^2-8\rho_1\rho_2\right)
$$

$\square$

**Proof of Theorem 3.1**

*Proof.* We have

$$
\begin{aligned}
\min_{l=0,\ldots,T}\left|\mathbb{E}\left[\|\nabla\mathcal{L}_S\left(\theta_l\right)\|\right]-\Delta\rho\beta\right| \leq & \frac{1}{T}\sum_{l=1}^{T}\left|\mathbb{E}\left[\|\nabla\mathcal{L}_S\left(\theta_l\right)\|\right]-\Delta\rho\beta\right| \\
\leq & \sqrt{\frac{1}{T}\sum_{l=1}^{T}\left(\mathbb{E}\left[\|\nabla\mathcal{L}_S\left(\theta_l\right)\|\right]-\Delta\rho\beta\right)^2} \\
\leq & \frac{2\left[\beta\Delta\left(\sigma^2+G^2\right)\right]^{1/4}}{T^{1/4}}+\beta\sqrt{6\left(7\rho_1^2+7\rho_2^2-8\rho_1\rho_2\right)}
\end{aligned}
$$

This implies that

$$
\begin{aligned}
\min_{l=0,\ldots,T}\mathbb{E}\left[\|\nabla\mathcal{L}_S\left(\theta_l\right)\|\right] \leq & \frac{2\left[\beta\Delta\left(\sigma^2+G^2\right)\right]^{1/4}}{T^{1/4}}+\beta\sqrt{6\left(7\rho_1^2+7\rho_2^2-8\rho_1\rho_2\right)} \\
& +\Delta\rho\beta.
\end{aligned}
$$

$\square$

## A.2 CONVERGENCE ANALYSIS

To do convergence analysis for Align-SAM, we make the following assumptions (Si & Yun, 2023):

**A1 ($G$-Lipchitz).** The loss function $\mathcal{L}_S$ is $G$-Lipchitz, i.e., $|\mathcal{L}_S(\theta)-\mathcal{L}_S(\theta')|\leq G\|\theta-\theta'\|$.

**A2 ($\beta$-smoothness).** The loss function $\mathcal{L}_S$ is $\beta$-smooth, if $\|\nabla\mathcal{L}_S\left(\theta\right)-\nabla\mathcal{L}_S\left(\theta'\right)\|\leq\beta\|\theta-\theta'\|$ for all $\theta,\theta'$.

**A3 (Bounded variance).** For any batch $B\sim S$, $\mathbb{E}_B\left[\|\mathcal{L}_B\left(\theta\right)-\mathcal{L}_S\left(\theta\right)\|^2\right]\leq\sigma^2$ for all $\theta$.

**Theorem 3.** *Assume that the loss function $\mathcal{L}_S$ satisfies the assumptions A1,A2, and A3, and $L^*=\inf_\theta\mathcal{L}_S(\theta)>-\infty$. Under Align-SAM, starting from $\theta_0$ and the learning rate $\eta=\frac{\sqrt{\Delta}}{\sqrt{\beta T(\sigma^2+G^2)}}$, we have*

$$
\frac{1}{T}\sum_{l=1}^{T}\left(\mathbb{E}\left[\|\nabla\mathcal{L}_S\left(\theta_l\right)\|\right]-\Delta\rho\beta\right)^2 \leq \frac{4\sqrt{\beta\Delta\left(\sigma^2+G^2\right)}}{\sqrt{T}}+6\beta^2\left(7\rho_1^2+7\rho_2^2-8\rho_1\rho_2\right),
$$

*where $\Delta\rho=\rho_1-\rho_2$ and $\Delta=\mathcal{L}_S(\theta_0)-L^*$.*

Table 6: Domain Generalization setting. All models are trained on ImageNet-1k and then evaluated on ImageNet-1k (clean validation set), Imagenet-R, and Imagenet-C datasets.

| Method | Top-1 Acc | | | Top-5 Acc | | |
|---|---|---|---|---|---|---|
| | Imagenet | ImageNet-R | ImageNet-C | Imagenet | ImageNet-R | ImageNet-C |
| **ResNet18 - Transfer Learning** | | | | | | |
| SAM | 70.52 | 34.18 | 48.27 | 89.60 | 52.82 | 72.17 |
| Align-SAM | **70.88** | **34.38** | **48.69** | **89.94** | **53.23** | **72.61** |
| **ResNet18 - From Scratch** | | | | | | |
| SAM | 62.46 | 25.86 | 32.96 | 73.15 | 43.09 | 55.72 |
| Align-SAM | **63.64** | **26.20** | **34.06** | **73.45** | **43.99** | **57.42** |

**Corollary 3.1.** *Under the assumptions as in Theorem 3, we have*

$$\min_{l=0,\ldots,T} \mathbb{E}\left[\|\nabla \mathcal{L}_S\left(\theta_l\right)\|\right] \leq \frac{2\left[\beta\Delta\left(\sigma^2 + G^2\right)\right]^{1/4}}{T^{1/4}} + \beta\left(\sqrt{6\left(7\rho_1^2 + 7\rho_2^2 - 8\rho_1\rho_2\right)} + \Delta\rho\right).$$

It is well known that the normalized (practical) version of SAM *does not converge* to the minimizer of the training loss, as rigorously demonstrated in Si & Yun (2023) (Theorem 4.6), one of the most comprehensive analyses of SAM's convergence behavior. Our proposed approach shares the *same convergence rate* as standard SAM, as established in Si & Yun (2023) (Theorem 4.6).

## A.3    ADDITIONAL EXPERIMENTS

**Experiments of Domain Generalization**    To strengthen the claims about Align-SAM, we evaluated Align-SAM's robustness under domain shifts by training on ImageNet-1K and testing on ImageNet-1K (clean test set), ImageNet-R (artistic renditions), and ImageNet-C (with various corruptions). These shifts demonstrate Align-SAM's robustness, as it consistently outperforms SAM across all setups. The results are shown in Table 6.

**Experiments of meta-learning setting**    The concept of Align-SAM is inspired by the agnostic approach in the MAML setting, where the meta-model is optimized on the meta-training set but aims to minimize loss on the validation set, assuming both the training and validation sets are from the same data distribution. Different from this original idea, Align-SAM uses the gradient from another auxiliary set as an indicator to close the generalization gap between the training and testing sets. Despite this difference, both approaches share the same underlying objective, making it reasonable to expect that applying Align-SAM in the MAML setting will result in improved generalization performance.

Table 7: Meta-learning results on Mini-Imagenet dataset. All baseline results are taken from Abbas et al. (2022)

| Method | Accuracy | |
|---|---|---|
| | 5 ways 1 shot | 5 ways 5 shots |
| MAML | 47.13 | 62.20 |
| SHARP-MAML$_{low}$ | 49.72 | 63.18 |
| Align-SAM$_{low}$ | **50.08** | **64.29** |

We compare our approach with standard MAML and Sharp-MAML (Abbas et al., 2022), which also address the loss landscape flatness in bilevel models. MAML is typically framed as a bilevel optimization problem, consisting of a meta-update step to learn a shared model initialization and a fine-tuning step to adapt task-specific models. Sharp-MAML analyzes the geometry of MAML's loss landscape and introduces the use of SAM to avoid sharp local minima in MAML loss functions. Sharp-MAML proposes three variants: Sharp-MAML$_{low}$ (applying SAM only to the fine-tuning step), Sharp-MAML$_{up}$ (applying SAM only to the meta-update step), and Sharp-MAML$_{both}$ (applying SAM to both steps). Since our Align-SAM shares the same objective as SAM to improve model

Table 8: Meta-learning results on Omniglot dataset. All baseline results are taken from Abbas et al. (2022)

| Method | Accuracy | |
|---|---|---|
| | 20 ways 1 shot | 20 ways 5 shots |
| MAML | 91.77 | 96.16 |
| SHARP-MAML$_{low}$ | **92.89** | 96.59 |
| Align-SAM$_{low}$ | 92.66 | **97.28** |

generalization it can replace SAM optimization in both the meta-update and fine-tuning steps of the MAML model.

The experiments follow the setup from Abbas et al. (2022), specifically using the Sharp-MAML$_{low}$ variation, which focuses on minimizing the sharpness of meta-models fine-tuned on the meta-training set. For Align-SAM, we set $\lambda = 2$ and follow the setup exactly for $\rho$, inner gradient steps, and step size in Abbas et al. (2022) for both Sharp-MAML$_{low}$ and Align-SAM$_{low}$. The results are reported in Tables 7 and 8. Our method outperforms most baselines with significant improvements, demonstrating the effectiveness of Align-SAM and its flexibility across various settings.

**Cosine similarity of gradients**   In Theorem 2, we prove that minimizing the loss function $\mathcal{L}_{B^t}$ could encourage two gradients $\nabla_\theta \mathcal{L}_{B^t}\left(\tilde{\theta}_l^t\right)$ and $\nabla_\theta \mathcal{L}_{B^a}\left(\tilde{\theta}_l^a\right)$ to be more congruent since our update aims to maximize its lower bound, which is $\nabla_\theta \mathcal{L}_{B^t}(\theta_l) \cdot \nabla_\theta \mathcal{L}_{B^a}\left(\tilde{\theta}_l^a\right)$. In this section, we measure the cosine similarity between two gradients $\nabla_\theta \mathcal{L}_{B^t}(\theta_l)$ and $\nabla_\theta \mathcal{L}_{B^a}\left(\tilde{\theta}_l^a\right)$ before (denoted as $cosine_b$ in Figure 1a) and after (denoted as $cosine_a$ in Figure 1b) updating the model and measure the change of these two score (denoted as $change$). Mathematically, we define $cosine_b = \frac{\nabla_\theta \mathcal{L}_{B^t}(\theta_l) \cdot \nabla_\theta \mathcal{L}_{B^a}\left(\tilde{\theta}_l^a\right)}{\|\nabla_\theta \mathcal{L}_{B^t}(\theta_l)\|_2 \|\nabla_\theta \mathcal{L}_{B^a}\left(\tilde{\theta}_l^a\right)\|_2}$, $cosine_a = \frac{\nabla_\theta \mathcal{L}_{B^t}(\theta_{l+1}) \cdot \nabla_\theta \mathcal{L}_{B^a}\left(\tilde{\theta}_{l+1}^a\right)}{\|\nabla_\theta \mathcal{L}_{B^t}(\theta_{l+1})\|_2 \|\nabla_\theta \mathcal{L}_{B^a}\left(\tilde{\theta}_{l+1}^a\right)\|_2}$, and $change = \frac{cosine_a - cosine_b}{cosine_a}$.

As shown in Figure 1c, both SAM and Align-SAM improve the similarity after updating the model, this improvement also increases across training epochs. However, the similarity score of our Align-SAM is always higher than SAM across the training process, both before and after updating the model. It is evident that our Align-SAM encourages gradients of the training subset and the auxiliary subset to be more similar during the training process.

**Batch size $|B^a|$ and complexity**   Our method, as presented in Algorithm 1, use a gradient on the auxiliary subset as a helper indicator to lead the model to wider local minima while maintaining low loss on a random subset from the same distribution, and the model should be updated mainly using training samples. Increasing the batch size $|B^a|$ could potentially increase performance and training time. In Table 9, we present the results of Align-SAM with various sizes $|B^a|$ of CIFAR-100 with Resnet32 while maintaining a fixed training batch size $|B^t| = 512$. We consider performance and training complexity to be the trade-off of Align-SAM and find that setting $|B^t| = 4|B^a|$ works well for all experiments.

Additionally, we examine different numbers of epochs with SAM, Align-SAM, and LookBehind-SAM Mordido et al. (2024). As presented in Table 9, Align-SAM outperforms both SAM and Lookbehind-SAM under the same training time.

**Sensitivity of trade-off coefficient $\lambda$**   Throughout this paper, we used a consistent setting of $\lambda = 2$, which is the trade-off coefficient for combining gradients from $B^t$ and $B^a$. While this hyperparameter could be optimized for each experiment individually, we find that this configuration delivers good performance across most experiments. By setting $\lambda > 1$, we ensure that the perturbed model prioritizes maximizing the loss on the training mini-batch $B^t$ rather than minimizing it on the auxiliary mini-batch $B^a$. This approach encourages the model to focus primarily on minimizing sharpness during the actual update step in Formula 6.

Table 9: Experiments on different sizes of auxiliary mini-batch with a fixed size of training mini-batch is 512 samples

| Method | Epoch | Auxiliary batch-size | Accuracy | Training time per epoch | Total |
|---|---|---|---|---|---|
| SAM | 200 | - | $70.31 \pm 0.233$ | 11s | 36.6m |
|  | 220 |  | $70.45 \pm 0.303$ | 11s | 40.3m |
|  | 250 |  | $70.06 \pm 0.078$ | 11s | 45.8m |
|  | 270 |  | $71.15 \pm 0.293$ | 11s | 49.5m |
| Align-SAM | 200 | 16 | $70.58 \pm 0.219$ | 11s | 37.5m |
|  |  | 32 | $71.07 \pm 0.172$ | 12s | 40.0m |
|  |  | 64 | $70.67 \pm 0.049$ | 13s | 43.3m |
|  |  | 128 | $71.21 \pm 0.056$ | 14s | 46.6m |
|  |  | 256 | $71.04 \pm 0.207$ | 15s | 50.0m |
| SAM | 220 | - | $70.45 \pm 0.303$ | 11s | 40.3m |
| Align-SAM | 170 | 128 | $70.83 \pm 0.209$ | 14s | 39.6m |

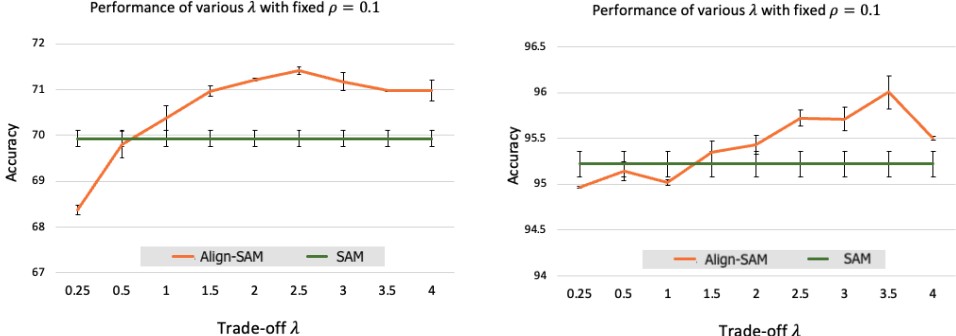

(a) Experiments with CIFAR-100 on Resnet32     (b) Experiments with Flower102 on EfficientNet-B2

Figure 2: Experiments of various trade-off $\lambda$ with fixed perturbation radius $\rho$

To verify the impact of this hyperparameter on model performance, we conduct experiments with varying values of trade-off $\lambda$ and present the results in Figure 2. Notably, the configuration where $\lambda \geq 1$ consistently yields higher accuracy compared to the setting where $\lambda < 1$. When decreasing $\lambda$, the model places more emphasis on minimizing the auxiliary loss, rather than sharpness on the training set during the actual update step in Formula 6, ultimately reducing performance.

**Analysis of loss landscape and eigenvalues of the Hessian matrix**   We demonstrate the effectiveness of Align-SAM in guiding models toward flatter regions of the loss landscape, as compared to both SAM and SGD, in Figures 3 and 4. The loss landscapes are visualized with the same setting, the blue areas represent lower loss values, while the red areas indicate higher loss values. Although SAM is shown to lead the model to a flatter region than SGD, Align-SAM achieves an even smoother and significantly flatter loss landscape, especially in experiments with EfficientNet-B2 in Figure 3.

To further validate that Align-SAM successfully locates minima with low curvature, we compute the Hessian of the loss landscape and report the five largest eigenvalues, sorted from $\lambda_1$ to $\lambda_5$, in Table 10. These eigenvalues provide insight into the curvature of the model at the optimized parameters. Larger eigenvalues indicate steeper curvature, meaning the model is more sensitive to small changes in its parameters. Conversely, smaller eigenvalues suggest flatter minima, which are typically associated with improved robustness, better generalization, and reduced sensitivity to overfitting. Negative eigenvalues indicate non-convex curvature in certain directions.

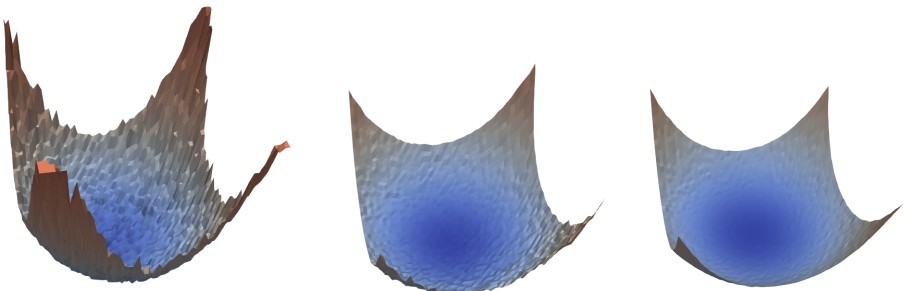

Figure 3: Loss landscape of **EffecientNet-B2** trained on Flower102 dataset with **(left)** SGD, **(middle)** SAM, and **(right)** Align-SAM.

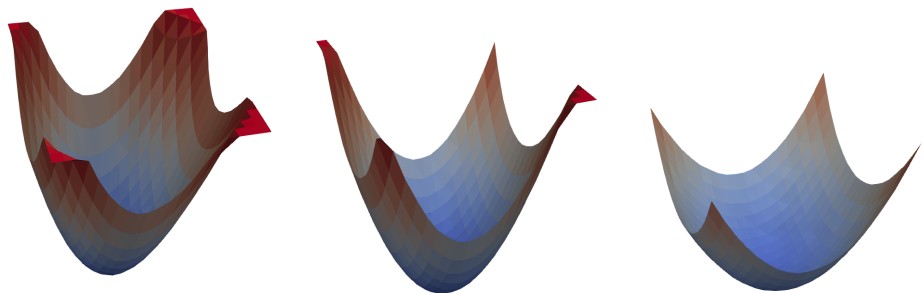

Figure 4: Loss landscape of **ResNet32** trained **(left)** SGD, **(middle)** SAM, and **(right)** Align-SAM on Cifar100 dataset.

| Methods | Ratio of top-5 largest eigenvalues of Hessian matrix | | | | |
|---|---|---|---|---|---|
| | $\lambda_1$ | $\frac{\lambda_1}{\lambda_2}$ | $\frac{\lambda_1}{\lambda_3}$ | $\frac{\lambda_1}{\lambda_4}$ | $\frac{\lambda_1}{\lambda_5}$ |
| **EfficientNet-B2 on Flower102** | | | | | |
| SGD | $2.05 \times 10^5$ | 4.55 | 7.88 | $-4.36$ | $-4.18$ |
| SAM | $1.61 \times 10^3$ | 1.27 | 1.31 | 1.54 | $-1.65$ |
| Align-SAM | $0.61 \times 10^3$ | 1.48 | 1.64 | 1.90 | 1.96 |
| **Resnet32 on Cifar100** | | | | | |
| SGD | $3.07 \times 10^5$ | 1.27 | 1.46 | 1.87 | 2.13 |
| SAM | $1.42 \times 10^5$ | 1.47 | 1.65 | 1.79 | 1.89 |
| Align-SAM | $1.11 \times 10^5$ | 1.32 | 1.68 | 1.85 | 2.09 |

Table 10: Eigenvalues of Hessian matrix

As shown in Table 10, Align-SAM consistently achieves positive and lower eigenvalues compared to the baseline methods, suggesting that it effectively leads the model toward flatter regions of the loss landscape. These results further support the efficacy of Align-SAM in optimizing for smoother and more stable solutions across a variety of architectures and tasks.

## A.4 ETHICAL STATEMENT AND THE USE OF LARGE LANGUAGE MODELS

We used a large language model (ChatGPT) to help with editing this paper. It was only used for simple tasks such as fixing typos, rephrasing sentences for clarity, and improving word choice. All ideas, experiments, and analyses were done by the authors, and the use of LLMs does not affect the reproducibility of our work.

