# OpenReview forum: "Align-SAM: Seeking Flatter Minima for Better Cross-Subset Alignment"
_ICLR.cc/2026/Conference — ICLR 2026 Poster_

### Official Review · Reviewer_B43K · 2025-10-17

**Soundness:** 3
**Presentation:** 3
**Contribution:** 3
**Rating:** 8
**Confidence:** 4

**Summary:**

The paper propses Align-SAM, a variation of SAM which modifies the SAM update to better align gradients between different subsets of the training set. The paper demonstates theoretically that indeed their algorithm leads to better alignment as well as the same convergence rate as SAM. Empirically, the method outperforms SAM variants.

**Strengths:**

The main strength of this paper in my view are the strong empirical results. Align-SAM's strong empirical performance is demonstrated on numerous datasets, architectures and against several baseliens. The method is also intuitive and is theoretically motivated.

**Weaknesses:**

The paper doesn't too have many weaknesses in my view. First, in Figure 2, the tradeoff coefficient range cuts off on the lower side at 0.25. Is it possible to consider a tradeoff coefficient of zero? Relatedly, is there a study varying the size of \rho? Second, several of the figures and tables are too small- I hope the authors use the extra page to enlarge these (e.g. Table 2, 3, 5, Figure 1).

Minor comments
- Is "Agnostic-SAM" in Figure 2 another name for Align-SAM?

**Questions:**

- What is the behavior of the method when the tradeoff coefficient is 0?
- Is there a study on how changing \rho affects performance?
- See minor comments above

---

> ### Author Response · Authors · 2025-11-21
> **Trade-off coeffient $\lambda=0$**
>
> Thank you for your comment. Below, we provide the results for the case $\lambda = 0$, where the perturbed model in formula (5) is computed without using the training set; the training mini-batches are still used only in the final update step. In this setting, the update rule in Equation (6) no longer incorporates sharpness information from the training mini-batches, which can naturally lead to reduced performance. For all experiments, we trained DenseNet-121 from scratch for 100 epochs on CIFAR-100, using $B^t = 128$ and $B^a = 32$. The corresponding results are reported in Table 1.
>
> _Table 1. Experiment with various values of λ_
> | **$\lambda$** | **Accuracy** |
> |----------------------|-----------------------|
> | 0                    | 66.72                 |
> | 0.1                  | 67.59                 |

---

> ### Author Response · Authors · 2025-11-21
> **Ablation study on perturbation radius $\rho$**
>
> We did not include the ablation study on $\rho$ in the original submission; however, we provide the results here and will update the paper accordingly. In SAM, its variants, and Align-SAM, $\rho$ controls the radius of the perturbation and therefore determines how far the perturbed model can move from the current parameters $\theta_t$. As expected, a too-large $\rho$ pushes the perturbation beyond the local region where the loss landscape is informative, causing the update to lose meaningful local curvature and geometric structure, while setting it too small reduces the method to behavior similar to SGD. This behavior has been analyzed extensively in prior SAM work and is consistent with the results we observe. In all of our experiments, we adopted the standard $\rho$ values commonly used in the SAM literature to ensure fair comparability and to isolate the contribution of our alignment mechanism.
>
> _Table 2. Ablation study on perturbation radius $\rho$_
>
> | $\rho$    | 0.2   | 0.17  | 0.15  | 0.13  | 0.1          | 0.07  | 0.05  | 0.035 | 0.025 | 0.01  | 0.005 |
> |-----------|-------|-------|-------|-------|--------------|-------|-------|-------|-------|-------|-------|
> | Align-SAM | 67.52 | 68.16 | 68.09 | 68.77 | **69.24** | 68.58 | 68.59 | 68.45 | 67.69 | 66.93 | 67.45 |
>
>
> &nbsp;
>
> ---
>
> **Typo and grammatical error** Thank you for pointing out the typos in our paper. The term "Agnostic-SAM" is a mistake and should be corrected to "Align-SAM." We will carefully proofread to avoid such errors.

---

> ### Author Response · Authors · 2025-11-26
> **Revised version is uploaded**
>
> Dear reviewer,
>
> We have uploaded a revised version of our paper that addresses all issues raised by the reviewers, including the related works, clarifications in the theorem, additional experimental results, and enlarged tables. All changes in both the main paper and the appendix are highlighted in violet color for your convenience. The ongoing experiments will be updated later.
>
> Thank you for taking the time to review our work.

---

### Official Review · Reviewer_2St2 · 2025-10-19

**Soundness:** 2
**Presentation:** 2
**Contribution:** 2
**Rating:** 4
**Confidence:** 4

**Summary:**

The paper proposes Align-SAM, a novel optimization method that extends SAM to further enhance generalization under different subsets. Align-SAM guides the model toward solutions that minimize sharpness and loss on a primary training subset while simultaneously maintaining low loss on an independently drawn auxiliary subset sampled from the same distribution. This approach views generalization through the lens of cross-subset alignment and is designed to find minima that are not only resilient to local perturbations but also robust against distributional variability in each training iteration. Empirical results demonstrate consistent performance gains over SAM and its variants across diverse and challenging settings.

**Strengths:**

1. **Theoretical Foundation and Methodological Proposal**: The authors approach generalization from a novel perspective, framing it as an alignment across random subsets drawn from the same data distribution. Building on this viewpoint, the methodology is supported by Theorem 1, an extension of the PAC-Bayes theorem, which establishes an upper bound on generalization loss using an independently drawn auxiliary subset. Furthermore, Theorem 2 provides theoretical justification that the proposed iterative update scheme (Equations 4–6) successfully maximizes the dot product of the gradients computed on the primary batch and the auxiliary batch, thereby encouraging these gradients to become more congruent.

2. **Extensive Experiments**: Align-SAM is evaluated extensively across diverse and challenging settings, demonstrating consistent effectiveness. The experiments include: training from scratch on large-scale datasets (ImageNet, Food101), mid-sized datasets (CIFAR-100/10), transfer learning using various architectures (ResNet, EfficientNet, ViT-Adapter-S), and evaluation of robustness against noisy labels (CIFAR-10/100). Additionally, domain generalization tests show robustness under domain shifts (ImageNet-R and ImageNet-C). Ablation studies reinforce the core mechanism, showing that Align-SAM achieves significantly flatter loss landscapes and lower Hessian eigenvalues compared to SAM and SGD.

**Weaknesses:**

# Major Concerns

> 1. **Insufficient Explanation of Preliminaries**:
> The theoretical foundation and the proposed Align-SAM heavily rely on familiarity with SAM, making the manuscript very difficult to follow for readers unfamiliar with SAM's specifics. Specifically, in Theorem 1, rather than stating "Under some mild conditions similar to SAM (Foret et al., 2021)," the specific conditions necessary for the theorem's validity should be explicitly mentioned, as is standard for a formal theorem.

> 2. **Necessity of the Auxiliary Batch Set**:
> Align-SAM introduces an auxiliary batch requiring additional computational overhead due to extra forward and backward passes compared to standard SAM. Even though authors report Table 7 to alleviate the issues with this, there is still a question for the efficiency  of Align-SAM (please refer to W3(b) for details).


> 3. **Questions Regarding Fair Comparison**:
>
>> a. **Baseline Explanation**: The experimental baselines require further description. The paper compares Align-SAM against VASSO, GSAM, and LookSAM. Information should be provided (briefly) on the core methods utilized by each of these techniques, and crucially, whether they require an auxiliary set. This information is necessary for the reader to assess the fairness of the comparison and the complexity trade-offs.
>
>> b. **Practicality of Training Time**: Table 7 reports training time overhead based on CIFAR100 experiments. However, the large-scale ImageNet experiments utilize batch sizes of $\|B_t \|=2048$ and $\|B_a \|=512$. Given the scale of ImageNet and the fourfold increase in batch sizes, it is likely that the actual training time overhead is significantly larger. This raises doubts about the practical relevance of Table 7's results for large-scale implementations like ImageNet.


# Minor Concerns
> 4. **The manuscript requires general refinement:**
>
>> a. **Reference citation formats appear incorrect** or duplicated in the "Related Works" section.
>
>> b. **Notation is sometimes omitted**, e.g. $L$ in Theorem 1.
>
>> c. In Line L308, the text refers to "Asgnostic-SAM." Please confirm if this is a **typo** and should be "Align-SAM" or "Align-ASAM".
>
>> d.  In the formulation of the Taylor expansion leading to Equation (7) (L235), the step often involves an equality (=) based on a first-order  (L221). It is suggested to clarify whether the equality should be changed to an approximation (≈), and if so, discussion regarding the magnitude of the error term should be included for completeness.

**Questions:**

1. Could you elaborate on the reasoning why, in the statement of Theorem 1 (L172), $$\mathcal{L}_{\mathcal{D}} (\theta^* | \mathcal{S}^{a})$$ is used as the objective to minimize (Equation 3), rather than $\mathcal{S})$.

2. In line with W3(a), could you briefly explain the core mechanisms of baseline methods and explicitly state whether they require an auxiliary set?

---

> ### Author Response · Authors · 2025-11-22
> **Theorem 1**
>
> **Conditions in Theorem 1**
> Our Theorem 1 is inspired by the theoretical analysis of SAM and the PAC-Bayesian generalization bound. The “mild conditions” referenced in Theorem 1 correspond to the standard assumptions used in SAM’s analysis (Foret et al., 2021):
>
> $$
> L_{D}(\theta)\leq E_{\epsilon_{i}\sim N(0,\rho)}L_{D}(\theta+\epsilon)
> $$
>
> We will revise the paper to clarify this point and correct the notation $L$ in Theorem 1 during rebuttal time.
>
> &nbsp;
>
> **Replace $S$ by $S^a$**
> In our theoretical setup, both $S^a$ and $S$ (or $S^t$) are i.i.d. subsets drawn from the same underlying data distribution $D$. Since $S$ and $S^t$ refer to the same available training set, optimizing $\mathcal{L}_D(\theta^* \mid S)$ is equivalent in expectation to optimizing $\mathcal{L}_D(\theta^* \mid S^a)$.
>
>  Theorem 1 is written using $S^a$ to form a smooth connection between the original formulation in Equation (1) and the bi-level objective in Equation (3), where the auxiliary set plays a central role. In other words, because both subsets are unbiased samples from the same distribution, substituting $S^a$ for $S$ does not change the objective in expectation but allows us to naturally introduce the auxiliary set into the theoretical development. Moreover, although our theory is presented for $S^a$, it can be generalized for any subset $S$ i.i.d sampled from $D$.

---

> ### Author Response · Authors · 2025-11-22
> **Necessity of the Auxiliary Batch Set**
>
> - **All baselines use the same training data, and Align-SAM does not introduce any external data.**
> In practice, the auxiliary mini-batch in Align-SAM is sampled from the same training set used by VASSO, GSAM, LookSAM, and other baselines. It is not drawn from an external dataset and therefore does not increase the total data requirement. The comparison is fully fair with respect to data usage.
>
> - **Several SAM-based baselines also incur additional computational cost.**
> Compared to the original SAM, methods such as  Lookbehind-SAM, and Lookahead-SAM require various extra gradient computations or additional ascent/descent steps. However, these methods use extra gradients in very different ways. VASSO, GSAM also require additional gradients compared to SAM in the update formula, but propose to reuse the gradients from previous steps so that the computational cost is similar to SAM. To the best of our knowledge, Align-SAM is the only method that computes its extra gradient on a separate auxiliary mini-batch. This is central to our method’s motivation: stabilizing the sharpness-aware perturbation by aligning curvature information across independent subsets.
>
> - **The auxiliary batch size is small and does not inflate the effective batch size.** As shown in Table 7, Align-SAM does not require a large auxiliary batch to be effective: Small auxiliary batches (16–64 samples) already yield stable gains. Larger auxiliary batches (e.g., 256) do not improve performance and may degrade it. This aligns with well-known results that large-batch training can harm generalization in SGD and SAM [1, 2, 3], due to reduced gradient diversity. Thus, the added computational cost is modest, and Align-SAM does not depend on inflating the effective batch size.
>
> [1] Keskar, Nitish Shirish, Dheevatsa Mudigere, Jorge Nocedal, Mikhail Smelyanskiy, and Ping Tak Peter Tang. "On large-batch training for deep learning: Generalization gap and sharp minima." arXiv preprint arXiv:1609.04836 (2016).
>
> [2] Luo, Haocheng, Mehrtash Harandi, Dinh Phung, and Trung Le. "Unveiling m-Sharpness Through the Structure of Stochastic Gradient Noise." arXiv preprint arXiv:2509.18001 (2025).
>
> [3] Wen, Kaiyue, Tengyu Ma, and Zhiyuan Li. "How sharpness-aware minimization minimizes sharpness?." In The Eleventh International Conference on Learning Representations. 2022.

---

> ### Author Response · Authors · 2025-11-22
> **Computational cost and batch size**
>
> In the ImageNet experiments, both Align-SAM and all baseline methods use the same training batch size of $|B^t| = 2048$. Align-SAM only adds an auxiliary mini-batch of $|B^a| = 512$, which does not change the total number of training iterations or epochs. Please refer to Table 1 below for the training time of the ImageNet-1K dataset. We chose this auxiliary batch size by following the recommended ratio $|B^t| = 4|B^a|$ in our CIFAR ablations to avoid hyperparameter tuning cost on large-scale datasets (which is really expensive for the ImageNet dataset). As shown in Table 7, even very small auxiliary batches already provide stable improvements with manageable computational overhead.
>
> For large datasets like ImageNet, we agree that the additional cost becomes more noticeable when accumulated over many iterations, as provided below. However, Table 7 illustrates the relative cost–benefit trade-off. In particular, Align-SAM trained for 200 epochs still outperforms SAM trained for 270 epochs, demonstrating that the alignment mechanism provides meaningful gains even under tighter computational budgets.
>
> &nbsp;
>
> _Table 1. Training time for experiments on ImageNet with $|B^t|=2048$ and $|B^v|=512$. ll experiments are conducted on
> the same single NVIDIA A100 GPU_
> | **Model** | **Method** | **Training time per iteration** | **Training time per epoch** |
> |-----------|------------|-----------------------------|-----------------------------|
> | ResNet18  | SAM        | 6.57s        |    68.5m                       |
> |           | Align-SAM  | 7.02s     |  73.2m                       |
> | ResNet34  | SAM        | 6.63s       |    69.2m                       |
> |           | Align-SAM  | 7.08s       |  73.9m                       |

---

> ### Author Response · Authors · 2025-11-22
> **Taylor expansion in Equation (7)**
>
> Thank reviewer for pointing out this error. We will edit our paper accordingly.
>
> In Equation (7), we omit the higher-order term because of the common assumption in SAM and its variance method that $O(|\tilde{\theta}^t_l - \theta_l|)$ is small by controlling a small $\rho$.

---

> ### Author Response · Authors · 2025-11-22
> **Briefly explain the core mechanisms of baseline methods**
>
> **VASSO:** The authors observe that large gradient variance harms generalization (Observation 2 in their paper). To mitigate this, instead of relying solely on the current gradient as SAM does, VASSO uses an exponential moving average of past and current gradients to produce a more stable perturbation direction.
>
> **GSAM:** GSAM improves sharpness-aware training by minimizing the surrogate gap, which directly corresponds to reducing sharpness. It combines the gradient at the current model (\theta_t) with the gap of gradients on the losses at the perturbed model (\tilde{\theta}_t) and at (\theta_t). GSAM also removes the descent-direction component of this gap that is aligned with the SAM ascent direction, resulting in a more stable and well-behaved update step.
>
> **LookSAM:** LookSAM analyzes the gap between the gradient at (\theta_t) and the gradient at the perturbed model (\tilde{\theta}_t), and proposes to reuse this gap for (k) steps to reduce SAM’s computational cost. When (k = 1), the method expects to match SAM computational cost and performance; when (k > 1), it expects to reduce computational overhead with only a small drop in performance.
>
> In all these methods, any additional gradients are computed on the **same training mini-batch** (or reused across iterations), not on an auxiliary set. Only Align-SAM introduces an auxiliary mini-batch to compute its extra gradient, which is central to our method’s motivation and distinction. However, it is important to note that the **auxiliary batch was also sampled from the same training dataset, without adding any external dataset.**

---

> ### Author Response · Authors · 2025-11-26
> **Revised version is uploaded**
>
> Dear reviewer,
>
> We have uploaded a revised version of our paper that addresses all issues raised by the reviewers, including the related works, clarifications in the theorem, additional experimental results, and enlarged tables. All changes in both the main paper and the appendix are highlighted in violet color for your convenience. The ongoing experiments will be updated later.
>
> Thank you for taking the time to review our work.

---

> > ### Comment · Reviewer_2St2 · 2025-11-27
> >
> > I appreciate the authors’ thorough and well-organized rebuttal. The responses effectively addressed my previous concerns.
> > Given these improvements, I am satisfied with the authors’ clarifications and am updating my overall assessment accordingly. I will raise my score to 6.
> > Thank you again for your careful and constructive engagement during the rebuttal process.

---

> ### Author Response · Authors · 2025-11-27
>
> We sincerely appreciate your time, your consideration and valuable comments that have helped us strengthen and improve our work.

---

### Official Review · Reviewer_Engx · 2025-10-31

**Soundness:** 3
**Presentation:** 3
**Contribution:** 3
**Rating:** 6
**Confidence:** 3

**Summary:**

This paper revisits Sharpness-Aware Minimization (SAM) from the perspective of cross-subset alignment. The authors argue that good generalization requires not only finding flat minima on the training subset, but also maintaining stable performance across independently sampled subsets from the same data distribution. To achieve this, they propose Align-SAM, which incorporates an auxiliary mini-batch and encourages gradient alignment between the primary and auxiliary subsets during training. The method is theoretically supported by a PAC Bayes generalization bound and a gradient congruence theorem. Extensive experiments validate the effectiveness of the proposed method.

**Strengths:**

- The paper presents a clear motivation. It aims to reduce sharpness and loss on the primary subset while maintaining low loss on the auxiliary subset, aligning well with the goal of improving generalization.
- The paper is theoretically grounded. It builds on PAC-Bayesian theory to formalize how cross-subset alignment reduces generalization error, and provides mathematical guarantees showing that the proposed update rule enhances gradient congruence across subsets, thereby offering a solid theoretical foundation for the method.
- The experiments span a wide range of tasks, including standard classification, transfer learning, and learning with label noise, demonstrating the robustness and effectiveness of the proposed method across diverse scenarios.

**Weaknesses:**

- According to Algorithm 1, the proposed method requires four backward passes for each parameter update. Although the authors state that using a much smaller auxiliary batch size $|B^a|$ than the target batch size $|B^t|$ can mitigate the computational overhead, this strategy may necessitate a large overall batch size (e.g., Table 7 uses a batch size of 512), and large-batch training may potentially harm performance.

- The experiments primarily evaluate CNN-based architectures (e.g., WideResNet, PyramidNet, DenseNet). Including results on transformer-based models such as ViT would further strengthen the paper, given their broad adoption and distinct optimization characteristics.

- More representative SAM-based methods, such as SAF [1], GAM [2], ImbSAM [3], CC-SAM [4], and Focal-SAM [5] should be included in the related work for a more comprehensive review.
- There are some typos, such as in line 307, where "Asgnostic-SAM" should be corrected to "Agnostic-SAM"

-----

[1] Sharpness-Aware Training for Free, NeurIPS 2022

[2] Gradient Norm Aware Minimization Seeks First-Order Flatness and Improves Generalization, CVPR 2023

[3] ImbSAM: A Closer Look at Sharpness-Aware Minimization in Class-Imbalanced Recognition, ICCV 2023

[4] Class-Conditional Sharpness-Aware Minimization for Deep Long-Tailed Recognition, CVPR 2023

[5] Focal-SAM: Focal Sharpness-Aware Minimization for Long-Tailed Classification, ICML 2025

**Questions:**

Please see above.

---

> ### Author Response · Authors · 2025-11-22
> **Batch-size and computational cost**
>
> It is correct that Algorithm 1 requires four backward passes in its full form. To keep the computation manageable, we recommend using a substantially smaller auxiliary batch size. As shown in Table 7, increasing the auxiliary batch size does not yield monotonic improvements. This is consistent with prior work showing that large-batch training can harm generalization in both SGD and SAM [1, 2, 3]. These results indicate that increasing the auxiliary batch alone, or increasing both the training and auxiliary batches, may actually degrade performance.
>
> Our ablation study suggests that a ratio of $|B^t| : |B^a| = 4 : 1$ provides a good balance between performance and computational efficiency. Moreover, $|B^a|$ can be reduced even further when computational resources are limited, while still maintaining the benefits of Align-SAM.
>
> [1] Keskar, Nitish Shirish, Dheevatsa Mudigere, Jorge Nocedal, Mikhail Smelyanskiy, and Ping Tak Peter Tang. "On large-batch training for deep learning: Generalization gap and sharp minima." arXiv preprint arXiv:1609.04836 (2016).
>
> [2] Luo, Haocheng, Mehrtash Harandi, Dinh Phung, and Trung Le. "Unveiling m-Sharpness Through the Structure of Stochastic Gradient Noise." arXiv preprint arXiv:2509.18001 (2025).
>
> [3] Wen, Kaiyue, Tengyu Ma, and Zhiyuan Li. "How sharpness-aware minimization minimizes sharpness?." In The Eleventh International Conference on Learning Representations. 2022.

---

> ### Author Response · Authors · 2025-11-22
> **Experiments on ViTs architecture**
>
> We agree that evaluating Align-SAM on transformer-based models such as ViT would further strengthen the paper, as these architectures exhibit distinct optimization behavior compared to CNNs. However, training ViTs from scratch is computationally prohibitive, especially when combined with SAM-style perturbation steps.
> Instead, to address this concern within reasonable compute limits, we evaluated Align-SAM under parameter-efficient fine-tuning using Adapters, as reported in Table 3. This setting allows us to examine transformer-based architectures while keeping the computational cost manageable. We believe these results provide meaningful evidence that Align-SAM is compatible with and beneficial for transformer-based models.
>
> _Table 3 in the main paper. Experiments of transfer learning setting on ImageNet with ViT_
> | **Model**     | **Top-1 Acc** |           | **Top-5 Acc** |            |
> |---------------|---------------|-----------|---------------|-----------|
> |               | SAM           | Align-SAM | SAM           | Align-SAM |
> | ViT-Adapter-S | 79.96         | **80.04** | 95.35         | **95.39** |

---

> ### Author Response · Authors · 2025-11-22
> **Related works, gramma error and typo**
>
> We thank reviewers for pointing out some related and interesting papers. We will revise and update our paper before the rebuttal time ends.

---

> ### Author Response · Authors · 2025-11-26
> **Revised version is uploaded**
>
> Dear reviewer,
>
> We have uploaded a revised version of our paper that addresses all issues raised by the reviewers, including the related works, clarifications in the theorem, additional experimental results, and enlarged tables. All changes in both the main paper and the appendix are highlighted in violet color for your convenience. The ongoing experiments will be updated later.
>
> Thank you for taking the time to review our work.

---

### Official Review · Reviewer_GCo1 · 2025-11-01

**Soundness:** 2
**Presentation:** 1
**Contribution:** 1
**Rating:** 2
**Confidence:** 4

**Summary:**

This paper proposes Align-SAM, a new optimization method based on Sharpness-Aware Minimization (SAM). The authors introduce the idea of "cross-subset alignment," arguing that good generalization requires a model trained on one subset of data to also perform well on an auxiliary, independently drawn subset. The proposed Align-SAM method extends SAM by optimizing for flat minima on a primary training batch while simultaneously enforcing low loss on this auxiliary batch. This is implemented as a modification to the SAM update rule that incorporates gradients from both batches, with the goal of encouraging gradient alignment. The authors provide empirical results across various settings, including image classification, transfer learning, and training with noisy labels, claiming consistent generalization improvements.

**Strengths:**

**Thorough Evaluation:** The paper presents a wide range of experiments, evaluating the method on standard classification, transfer learning, domain generalization, and noisy labels.

**Weaknesses:**

- **Marginal Gains:** The central claim of "consistent improvement" is overstated. In many key experiments (e.g., CIFAR classification and transfer learning), the performance gains over strong baselines like SAM and ASAM are marginal and often fall within the reported standard deviation. These minor benefits do not appear to justify the significant computational overhead of an extra forward/backward pass per step.
- **Lack of Novelty and Missing Discussion:** The novelty of the update rule is questionable, as it resembles previous works like Lookahead SAM [1] and Lookbehind SAM [2]. The authors fail to adequately discuss or compare against these highly relevant methods in the main text, making it difficult to assess the originality and relative performance of the proposed contribution.

[1] Yu, Runsheng, Youzhi Zhang, and James Kwok. "Improving sharpness-aware minimization by lookahead." *Forty-first International Conference on Machine Learning*. 2024.

[2] Mordido, Gonçalo, et al. "Lookbehind-SAM: k steps back, 1 step forward." *arXiv preprint arXiv:2307.16704* (2023).

**Questions:**

1. Should the authors clarify that this paper focus on *in-distribution* generalization, given that *out-of-distribution* generalization is a distinct concept?
2. What are the meaning of $\nabla\_\theta \mathcal{L}\_{B\_t}(\tilde{\theta}\_l^t)$ and $\nabla\_\theta \mathcal{L}\_{B\_a}(\tilde{\theta}\_l^a)$? Do we want to align the gradient of $B\_t$ and $B\_a$ at $\theta\_l$, i.e., $\nabla\_\theta \mathcal{L}\_{B\_t}({\theta}\_l)$ and $\nabla\_\theta \mathcal{L}\_{B\_a}({\theta}\_l)$, instead?
3. The comparison appears to be unfair because Align-SAM effectively operates with a larger batch size than the baseline methods.
4. Could you please add standard deviations to Table 1 and Table 3 for a more rigorous comparison? Also, given its importance, should the analysis of Figure 1 (currently in Appendix A.3) be moved to the main text?
5. How does Align-SAM compare to a simpler baseline that adds a direct (weighted) gradient alignment regularizer to the standard SAM loss?
6. Could the authors clarify the intuition for why Align-SAM improves performance under noisy labels, especially when the test dataset follows a different distribution?
7. Missing related work about SAM
- Yu, Runsheng, Youzhi Zhang, and James Kwok. "Improving sharpness-aware minimization by lookahead." Forty-first International Conference on Machine Learning. 2024.
- Nguyen, Tuan H., et al. "Changing the training data distribution to reduce simplicity bias improves in-distribution generalization." Advances in Neural Information Processing Systems 37 (2024): 68854-68896.

---

> ### Author Response · Authors · 2025-11-22
> **Marginal Gains in experiments with CIFAR classification and transfer learning**
>
> We thank the reviewer for raising this concern. We acknowledge that Align-SAM does not produce large gains in every setting, such as CIFAR-10 and Transfer learning on small datasets; however, our Align-SAM gains significantly on experiments with ImageNet training from scratch (Table 1) and Noisy labels (Table 5). Moreover, our proposed approach can be applied to a wide range of settings as we demonstrate in our experiments.
>
> On clean and well-behaved benchmarks such as CIFAR classification and standard transfer learning, the optimization landscape is already close to saturated. For instance, Lookahead-SAM achieves 0.2–0.4\% improvement on CIFAR-10 and 0.3–1\% on CIFAR-100 using CNN models, despite also requiring extra gradient computations. Even SAM with doubling the forward/backward steps, yields only modest improvements over SGD in these settings. This highlights that the limited performance gains that arise from the saturation of these benchmarks do not undermine the contribution and novelty of our method.
>
> We note that if following our practical recommendation of setting $|B^t| : |B^a| = 4:1$, the training of our AlignSAM is only slightly slower than SAM. Particularly, by setting $|B^t|= 512, |B^a| = 128|$ on CIFAR 100 with ResNet32, the training time per epoch of SAM is 11s (accuracy is 70.31%, while ours is 14s (accuracy is 71.21%). Please refer to Table 7 in Appendix A3 for more details.

---

> ### Author Response · Authors · 2025-11-22
> **Lack of Novelty and Missing Discussion**
>
> We appreciate the reviewer’s attention to related methods such as Lookahead-SAM and Lookbehind-SAM. While these approaches also introduce additional gradient steps relative to standard SAM, **their motivations, formulations, and update mechanisms differ substantially from ours**. We agree that the main paper should have emphasized this distinction more clearly.
>
> However, **we respectfully disagree with the suggestion that Align-SAM is an ensemble or restatement of Lookahead-SAM or Lookbehind-SAM.** Although all three methods require extra-gradient computations, the underlying problems they aim to solve,  their update formulas, and the motivations stand from different viewpoints.
>
> Lookahead-SAM and Lookbehind-SAM are both derived from the min–max formulation of the SAM objective and are explicitly inspired by the Lookahead mechanism. Lookahead-SAM uses extra gradients on the same training mini-batch $B_t$ to simultaneously update the perturbation vector $\epsilon_t$ and a “lookahead” model $W_t$, creating a forward-looking trajectory that avoids saddle points and converges to a stationary point under smoothness and bounded-gradient assumptions. To mitigate the computational problem of Lookahead-SAM, the authors proposed additional efficient methods that reuse and modify gradients in previous iterations. Lookbehind-SAM instead reuses several past models and their ascent directions (all computed on the same mini-batch $B^t$), averaging these historical ascent steps to stabilize the update. Both variants use extra gradients solely on the current mini-batch and solely to modify the trajectory (ahead or behind) of the optimization process. This is fundamentally different from our approach.
>
> **Align-SAM, in contrast, addresses a distinct problem and provides a new perspective on sharpness-aware optimization.** Rather than modifying the trajectory of the model, Align-SAM uses gradients computed on an auxiliary mini-batch $B^a$ that is independent of the current training batch $B^t$. The goal is to align curvature-sensitive directions across subsets—not to look ahead or behind the update trajectory in parameter space. Our extra-gradient step, therefore, addresses a different problem: cross-subset alignment of sharpness-aware perturbations in two different loss landscapes, enabling more stable perturbations under noisy labels, small-batch noise, and distributional drift. Align-SAM does not perform weight interpolation, trajectory smoothing, or backward/forward stepping, but it introduces a new mechanism that is not considered in either Lookahead-SAM or Lookbehind-SAM.
>
> **Novelty**: Our Align-SAM follows agnostic thinking as shown in Eq. (1). The central contribution is fundamentally different from Lookahead-based works. Specifically, our core idea is fundamentally novel: we investigate a model’s generalization by optimizing it primarily on the training set while encouraging strong performance on another subset from the same distribution. This naturally leads to a bi-level optimization problem, as formulated in Eq. (1) or its relaxed version in Eq. (3). The update rule is therefore designed to modify the sharpness-aware perturbation so that it reflects directions that improve both subsets—not merely the current mini-batch $B^t$ like existing SAM and its variances.

---

> ### Author Response · Authors · 2025-11-22
> **In-distribution and out-of-distribution**
>
> Our method is motivated by the distributional shift problem that naturally arises in real-world problems. In practice, both $B^t$ and $B^a$ are drawn from the same training distribution, **so the main focus of our work is indeed in-distribution generalization.** However, the stability introduced by Align-SAM, especially under noisy labels and mini-batch variability, can also improve robustness under mild distribution mismatch. As shown in Table 6 in the Appendix, when trained on ImageNet-1K and tested on three different distributions—ImageNet-1K (clean test set), ImageNet-R (artistic renditions), and ImageNet-C (corruptions). ImageNet-R and ImageNet-C are standard out-of-distribution benchmarks relative to ImageNet-1K. Across all these settings, Align-SAM consistently outperforms SAM, demonstrating that the alignment mechanism not only enhances in-distribution generalization but also provides robustness under distributional shift.
>
> Our method is motivated by the distributional shift problem that naturally arises in real-world problems. In practice, both $B^t$ and $B^a$ are drawn from the same training distribution, **so the main focus of our work is indeed in-distribution generalization.** However, the stability introduced by Align-SAM, especially under noisy labels and mini-batch variability, can also improve robustness under mild distribution mismatch. As shown in Table 6 in the Appendix, when trained on ImageNet-1K and tested on three different distributions—ImageNet-1K (clean test set), ImageNet-R (artistic renditions), and ImageNet-C (corruptions). ImageNet-R and ImageNet-C are standard out-of-distribution benchmarks relative to ImageNet-1K. Across all these settings, Align-SAM consistently outperforms SAM, demonstrating that the alignment mechanism not only enhances in-distribution generalization but also provides robustness under distributional shift.
>
> _Table 6 in the main paper. Domain Generalization setting. All models are trained on ImageNet-1k and then evaluated on ImageNet-1k (clean validation set), Imagenet-R, and Imagenet-C datasets._
> | **Method**                               |  | **Top-1 Acc**            |     |     | | **Top-5 Acc**   |          |
> |-----------------------------------------------------------|-----------------------------------|-----------------|----------------|------|-----------------------------------|----------------|----------------|
> |                                                           | Imagenet                          | ImageNet-R      | ImageNet-C     |      | Imagenet                          | ImageNet-R     | ImageNet-C     |
> |**ResNet18** |         **- Transfer**                          | **Learning**                |                |
> | SAM                                                       | 70.52                             | 34.18           | 48.27          |      | 89.60                             | 52.82          | 72.17          |
> | Align-SAM                                                 | **70.88**                   | **34.38** | **48.69**|      | **89.94**                   | **53.23**| **72.61**|
> | **ResNet18**     |    **- From**                               |  **Scratch**               |                |      |
> | SAM                                                       | 62.46                             | 25.86           | 32.96          |      | 73.15                             | 43.09          | 55.72          |
> | Align-SAM                                                 | **63.64**                   | **26.20** | **34.06**|      | **73.45**                   | **43.99**| **57.42**|
>
> &nbsp;
>
> Furthermore, we analyze Align-SAM in the meta-learning setting, where there are two separate meta-training and validation sets, both of which are involved in the training process. In this setting, the meta-model is optimized on the meta-training set but aims to minimize the loss on the validation set. We compare our approach with standard MAML [1] and Sharp-MAML [2], which also tackle loss landscape flatness in bi-level models. The results, as presented in Tables 1 and 2, indicate that our method consistently outperforms most baselines with significant improvements, demonstrating the effectiveness and flexibility of Agnostic-SAM across various settings.
>
>
> _Table 1. Meta-learning results on the Mini-Imagenet dataset. All baseline results are taken from [2]_
> | **Method**          | **Accuracy**  |           |
> |---------------------|---------------|----------------|
> |                     | **5 ways 1 shot** | **5 ways 5 shots** |
> | MAML [1]            | 47.13         | 62.20          |
> | SHARP-MAML [2]      | 49.72         | 63.18          |
> | Align-SAM           | **50.08**     | **64.29**      |

---

> > ### Author Response · Authors · 2025-11-23
> > **Additional experiments of meta-learning setting**
> >
> > _Table 2. Meta-learning results on the Omniglot dataset. All baseline results are taken from [2]_
> > | **Method**          | **Accuracy**  |           |
> > |---------------------|---------------|----------------|
> > |                     | **20 ways 1 shot** | **20 ways 5 shots** |
> > | MAML [1]            | 91.77          | 96.16          |
> > | SHARP-MAML [2]      | **92.89**        | 96.59          |
> > | Align-SAM           | **92.89**     | **97.28**      |
> >
> >
> > &nbsp;
> >
> > [1] Chelsea Finn, Pieter Abbeel, and Sergey Levine. Model-agnostic meta-learning for fast adaptation of deep networks. In International conference on machine learning, pages 1126–1135. PMLR,2017.
> >
> > [2] Abbas, Momin, Quan Xiao, Lisha Chen, Pin-Yu Chen, and Tianyi Chen. "Sharp-maml: Sharpness-aware model-agnostic meta learning." In International conference on machine learning, pp. 10-32. PMLR, 2022.

---

> ### Author Response · Authors · 2025-11-22
> **A simpler baseline that adds a direct (weighted) gradient alignment regularizer to the standard SAM loss**
>
> We interpret the reviewer’s suggestion as adding a direct gradient-alignment regularizer of the form $L_{\text{reg}} = 1 - \cos\big(\nabla_{\theta} L_{B^t}(\tilde{\theta}^t), \nabla_{\theta} L_{B^a}(\tilde{\theta}^a)\big)$
> and jointly minimizing this term alongside the SAM objective. While this is a natural idea in principle, it is extremely difficult to implement in practice. Minimizing $L_{\text{reg}}$ requires differentiating through the gradients themselves, which introduces second-order derivatives (Hessian–vector products) with respect to both $B^t$ and $B^a$. These computations are prohibitively expensive for large deep networks and standard training pipelines, and we are not aware of any existing method that performs such full gradient-alignment regularization efficiently without severe approximation or instability. **To our knowledge, no prior SAM-based work or regularization method has proposed a tractable approximation for this specific objective. If the reviewer could recommend an efficient implementation or a relevant reference, we would be happy to examine it.**

---

> ### Author Response · Authors · 2025-11-22
> **Why Align-SAM improves performance under noisy labels**
>
> A key observation for datasets containing both clean and noisy labels is that the gradient computed on the full training set, or on any mixed mini-batch, **is still dominated by the clean portion**. In practice, although some labels are corrupted, the aggregated gradient direction remains substantially closer to the clean-label gradient than to the gradient induced solely by the noisy subset (Table 2). This explains why standard SGD can still learn reasonable decision boundaries when the noise rate is not excessively high. **SAM and its variants further amplify this behavior, and Align-SAM explicitly strengthens it by additionally incorporating gradients from an auxiliary batch.**
>
> _Table 2. Let (A) denote the set of cleanly labeled samples, (B) the set of mislabeled samples, and (B') the same inputs as (B) but assigned their correct labels. We report the cosine similarity of gradient on each pair when training with the mixed set [A, B] using the CIFAR-10-noisy dataset on ResNet32._
>
> | **Noisy rate** | **Method** |                    | **Cosine similarity** |                        |          | **Gradient** |  **norm**     |           |
> |----------------|--------|---------|--------------|--------------|----------|--------|-------|-----------|
> |                |            | **$[A, B]$ and $[A]$** | **$[A, B]$ and $[B]$**    | **$[A, B]$ and $[A, B']$** | **$[A, B]$** | **$[A]$**             | **$[B]$** | **$[A, B']$** |
> | 0.2            | SGD        | 0.8917             | 0.0032                | 0.8164                 | 10.78    | 13.80             | 19.69 | 13.24     |
> |                | SAM        | 0.9333             | -0.0428               | 0.8912                 | 10.47    | 13.48             | 16.56 | 12.99     |
> |                | Align-SAM  | **0.9403**   | **-0.106**      | **0.9027**       | 9.79     | 12.69             | 15.09 | 9.79      |
> | 0.4            | SGD        | 0.8155             | 0.0085                | 0.6919                 | 7.56     | 13.59             | 12.41 | 13.53     |
> |                | SAM        | 0.8771             | -0.0347               | 0.7967                 | 6.82     | 11.58             | 9.46  | 10.90     |
> |                | Align-SAM  | **0.8852**   | **-0.073**      | **0.8185**       | 5.77     | 9.69              | 8.28  | 9.02      |
> | 0.6            | SGD        | 0.7442             | 0.0870                | 0.5777                 | 5.89     | 13.95             | 6.42  | 12.81     |
> |                | SAM        | 0.8248             | 0.0075                | 0.7186                 | 4.08     | 9.47              | 4.60  | 8.63      |
> |                | Align-SAM  | **0.8429**   | **-0.032**      | **0.7576**       | 4.12     | 10.23             | 3.82  | 9.33      |
> | 0.8            | SGD        | 0.5396             | 0.3064                | 0.3803                 | 1.26     | 4.80              | 1.64  | 4.44      |
> |                | SAM        | 0.6254             | 0.1967                | 0.5330                 | 0.84     | 3.48              | 0.95  | 3.25      |
> |                | Align-SAM  | **0.6372**   | **0.1252**      | **0.5643**       | 0.53     | 2.26              | 0.64  | 2.17      |
>
> Even at high noise levels, the gradient of the mixed dataset ([A, B]) remains much closer to the clean gradient on (A) than to the noisy gradient on (B). Its similarity to ([A, B']) further shows that the clean-consistent samples still dominate the aggregated direction.
>
> For SAM and its variants, this effect becomes even stronger. The sharpness-aware perturbation favors gradient directions that are consistently supported across samples, so the **high-variance, inconsistent gradients from noisy labels are naturally suppressed**. Empirically, we also observe that the gradient norm of the noisy subset (B) decreases significantly as the noise rate increases, and remains much smaller than that of ([A, B]), (A), and ([A, B']). This confirms that mislabeled samples have a limited impact on the SAM perturbation. Prior work similarly shows that SAM tends to bias training toward flatter, label-consistent regions, reducing the influence of noisy gradients; see [3] for a detailed mechanistic analysis.
>
> Align-SAM further enhances this robustness. By computing the perturbation using an auxiliary batch, it explicitly decorrelates the ascent direction from label-specific noise in the current batch. This **encourages the perturbed model to move in directions where gradients from both the training and auxiliary batches align—directions dominated by clean samples.** As a result, Align-SAM produces updates that better reflect the underlying clean-label structure, improving robustness to label noise and yielding more reliable generalization under mild distribution shifts.
>
> [3] Baek, Christina, Zico Kolter, and Aditi Raghunathan. "Why is SAM Robust to Label Noise?." The Twelfth International Conference on Learning Representations (2024).

---

> ### Author Response · Authors · 2025-11-22
> **Missing related works**
>
> We thank reviewers for pointing out some related and interesting papers. We will revise and update our paper before the rebuttal time ends.

---

> ### Author Response · Authors · 2025-11-22
> **$\nabla_{\theta} L_{B^t}(\tilde{\theta}^t), \nabla_{\theta} L_{B^a}(\tilde{\theta}^a)$ and the alignment of training and auxiliary batch**
>
> $\nabla_{\theta} L_{B^t}(\tilde{\theta}^t)$ denotes the gradient of the perturbed model (inner-max)  computed on the training batch $B^t$, and $\nabla_{\theta} L_{B^a}(\tilde{\theta}^a)$ denotes the corresponding gradient on the auxiliary batch $B^a$. SAM and its variants typically use each of these gradients independently to update the model (in two iterations). In contrast, our goal is to encourage alignment between the gradients from $B^t$ and $B^a$ not only at the current parameters $\theta_l$ but mainly at the actual update models. This ensures that the updated parameters $\theta_{l+1}$ lie in a region that is both aligned and flat with respect to the two subsets, ultimately improving generalization. We further provide a theoretical result showing that the dot product between these gradients is lower-bounded and becomes more congruent under Align-SAM in Theorem 2.

---

> ### Author Response · Authors · 2025-11-22
> **Batch size and the fair comparison**
>
> All experiments of **both baselines and Align-SAM use exactly the same batch size $|B^t|$**, so the **comparison is fair**. Moreover, simply increasing training time, performing extra gradient computations, or enlarging the batch size does not inherently improve performance. In fact, both SGD and SAM often degrade performance as batch size grows due to reduced gradient diversity and poorer sharpness exploration [4] [5] [6]. Align-SAM improvements do not benefit from a specific batch-size setting, but from its alignment-based perturbation mechanism, which stabilizes the sharpness-aware update rather than altering the batch-size regime.
>
>
> Additionally, we conducted additional experiments with varying batch sizes and training epochs. Specifically, we trained SAM with a batch size of 160 (with 200 and 250 epochs), and compared it to Align-SAM, which used a training batch size of 128 and a validation batch size of 32 (for 200 epochs). The results indicate that training SAM with larger batch sizes does not necessarily lead to improved performance (Note that we do not intend to conduct an intensive ablation study on SAM's behavior). Furthermore, training SAM for 250 epochs does not outperform Align-SAM, which was trained for 200 epochs (with the same total training time).
>
> _Table 3. Train SAM on WideResNet28-10 with different batch sizes and epochs._
>
> | Train batch size	| Num epochs |	Accuracy|
> |---|---|---|
> | 128| 	200	| 82.77|
> | 128| 	250	| 82.94|
> | 160| 	200	| 82.44|
> | 160| 	250	| 82.73|
> | 200| 	200	| 82.39|
> 200	| 250	| 82.32|
>
> &nbsp;
>
> _Table 4. Train Align-SAM on WideResNet28-10 with different batch sizes._
>
> | Train batch size| 	validation batch size	| Accuracy|
> |---|---|---|
> 128	| 32	| 83.24|
> | 128| 	64| 	83.49|
> | 128	| 128	| 83.72|
> | 128	| 256| 	83.60|
>
> &nbsp;
>
> [4] Keskar, Nitish Shirish, Dheevatsa Mudigere, Jorge Nocedal, Mikhail Smelyanskiy, and Ping Tak Peter Tang. "On large-batch training for deep learning: Generalization gap and sharp minima." arXiv preprint arXiv:1609.04836 (2016).
>
> [5] Luo, Haocheng, Mehrtash Harandi, Dinh Phung, and Trung Le. "Unveiling m-Sharpness Through the Structure of Stochastic Gradient Noise." arXiv preprint arXiv:2509.18001 (2025).
>
> [6] Wen, Kaiyue, Tengyu Ma, and Zhiyuan Li. "How sharpness-aware minimization minimizes sharpness?." In The Eleventh International Conference on Learning Representations. 2022.

---

> ### Author Response · Authors · 2025-11-22
> **Standard deviations of experiments on ImageNet-1K dataset**
>
> Experiments on ImageNet-1K are significantly longer than those on other datasets, so within the rebuttal period, we provide additional results in the **transfer-learning** setting instead (Table 3 in the main paper). We also include new experiments of **Lookbehind-SAM** on ResNet34 using $k = 2$ and $\alpha = 0.8$. All experiments use the same hyperparameters: training batch size $|B_t| = 2048$, fine-tuning for 50 epochs, and perturbation radius $\rho = 0.05$. The results are summarized in Table 5 below.
>
> &nbsp;
>
> _Table 5. Transfer learning on ImageNet with ResNet models._
>
> | Model |                  | Top-1 Acc |                            |                  | Top-5 Acc |                            |
> |------------------------|------------------|-------------------------------|----------------------------|------------------|-------------------------------|----------------------------|
> |                        | SAM              | Lookbehind-SAM                | Align-SAM                  | SAM              | Lookbehind-SAM                | Align-SAM                  |
> | Resnet18               | 70.44 $\pm$ 0.12 | -                             | **70.92 $\pm$ 0.05**    | 89.63 $\pm$ 0.04 | -                             | **89.90  $\pm$ 0.04** |
> | Resnet34               | 73.40 $\pm$ 0.48 | 73.50 $\pm$ 0.09              | **73.94 $ \pm$ 0.14** | 91.43 $\pm$ 0.19 | 91.52 $\pm$ 0.03              | **91.78 $\pm$ 0.03** |

---

> ### Author Response · Authors · 2025-11-26
> **Revision version is uploaded**
>
> Dear reviewer,
>
> We have uploaded a revised version of our paper that addresses all issues raised by the reviewers, including the related works, clarifications in the theorem, additional experimental results, and enlarged tables. All changes in both the main paper and the appendix are highlighted in violet color for your convenience. The ongoing experiments will be updated later.
>
> Thank you for taking the time to review our work.

---

### Author Response · Authors · 2025-12-01
**Summarize the discussion**

Dear Area Chair,

Thank you for taking the extra time to review the rebuttal process of our paper. I would like to briefly summarize the discussion during the rebuttal period.

### **Summarize the discussion:**
- Only reviewer 2St2 was active in the discussion (before the system incident happened) and was **fully satisfied and raised their score.**

- We provided complete rebuttals to all questions from reviewer GCo1, and we respectfully emphasize that the novelty concerns are from a misunderstanding of the conceptual difference between Align-SAM and Lookahead-SAM and Lookbehind-SAM (both baselines start from the min-max problem). **Our theoretical formulation, auxiliary-batch mechanism, and cross-subset alignment perspective are all distinct and original.**

- **Reviewers Engx and B43K were already positive**

- The revised paper addressed all clarifications, corrected notation, expanded related work, and new experiments (highlighted in violet in the updated PDF).

---
&nbsp;

### **Detail discussion:**

&nbsp;

**1. Reviewer GCo1 — We respectfully disagree with the “lack of novelty” claim**

- Reviewer GCo1 (gave 2 rating score) raised concerns mainly about novelty and experimental marginal gains. We carefully clarified that Align-SAM introduces a fundamentally different mechanism from Lookahead-SAM and Lookbehind-SAM. [Detail here](https://openreview.net/forum?id=LvllbDxKZt&noteId=h2LO22Vmug)

- We also provided expanded related work, detailed clarifications and analysis, and added new experiments ( with the ImageNet-1K dataset and also new settings) within the rebuttal period.

While the reviewer raised numerous questions about baselines, batch-size fairness, and novelty, we addressed all of them thoroughly. The reviewer did not provide further follow-up after our rebuttal (likely due to the system issue), so we do not know whether or not the reviewer GCo1 was satisfied. However, we believed that our responses directly and comprehensively resolve their points.

&nbsp;

**2. Reviewer 2St2 — fully satisfied and raised score**

Reviewer 2St2 (gave 4 rating score initially, then raised to 6) raised several questions regarding the theoretical assumptions, necessity of the auxiliary batch, fairness of comparisons with baselines, and the nature of the Taylor approximation. We provided comprehensive responses addressing each point (Theorem 1 conditions, auxiliary-batch rationale, baseline mechanisms, computational cost, and notation corrections). After reviewing our rebuttal and the revised manuscript, **Reviewer 2St2 expressed full satisfaction and explicitly increased their score to 6.** (before system issue)

&nbsp;

**3. Reviewers Engx and B43K were already positive**

- Reviewer Engx: Score 6, positive assessment of theory and experiments, with only minor concerns (typos, related work expansion).
- Reviewer B43K: Score 8, strongly positive overall, and request additional ablation studies to strengthen our work.

&nbsp;


We kindly hope that the Area Chair will consider this when making the final decision.

Thank you very much for your time and for coordinating the review process.

Sincerely,

---

### Meta-Review · Area_Chair_TVgL · 2026-01-05

**Summary:**

This paper focuses on improving generalization by proposing a principled extension of sharpness-aware minimization. The initial review scores are 2, 4, 6, and 8. After the rebuttal and discussion, one reviewer raises their score from 4 to 6, while the others maintain their original evaluations. Overall, the reviewers agree that the paper provides a technically sound approach, supported by empirical results. Based on the reviews and the discussion, the reviewers鈥?opinions can be summarized as follows:

- The proposed method is well motivated and has a clear connection to optimization and generalization theory.
- The theoretical analysis supports the core idea, and the rebuttal clarifies earlier confusion about assumptions and mechanisms.
- The experimental evaluation is broad and convincing, covering standard classification, transfer learning, noisy-label settings, and meta-learning.
- The method shows consistent improvements over strong baselines, and the empirical results are robust across different datasets and training settings.
- The method introduces additional computational overhead compared to standard training, which may limit scalability in very large-scale scenarios.

Additionally, Reviewer GCo1 argues in the initial review that the novelty may appear incremental compared to prior sharpness-aware methods. In the rebuttal, the authors analyze the differences between this work and related methods (Lookahead-SAM and Lookbehind-SAM). However, Reviewer GCo1 does not join the discussion and does not respond to the rebuttal. After reading the rebuttal carefully, I acknowledge the novelty of this paper in its theoretical formulation, auxiliary-batch mechanism, and cross-subset alignment perspective.

**Reviewer Concerns:**

Reviewer GCo1 raises concerns primarily about novelty, clarity of the theoretical contribution, and the relationship to prior work. After carefully reading the rebuttal, I acknowledge the novelty of this paper in its theoretical formulation, auxiliary-batch mechanism, and cross-subset alignment perspective.
- Reviewer 2St2 expresses concerns regarding theoretical clarity, fairness of comparisons, and computational cost. The authors respond comprehensively by clarifying assumptions, correcting notation, expanding theoretical explanations, and providing additional experiments. This reviewer states that the concerns are addressed and increases their score accordingly.
- Reviewer Engx is generally positive, with minor concerns about presentation and computational aspects. These issues are addressed in the rebuttal.
- Reviewer B43K provides a strong positive assessment focused on empirical results, with only minor clarification questions, all of which are addressed by the authors.

Overall, the authors’ rebuttal addresses the concerns raised by the reviewers.

**Reviewer Scores:**

- Reviewer GCo1 does not participate in the discussion and is likely to maintain the original score of 2.
- Reviewer 2St2 explicitly acknowledges that the concerns are addressed and increases the score from 4 to 6.
- Reviewer Engx does not participate further and is likely to maintain the score  6.
- Reviewer B43K maintains a positive stance and is likely to keep the score 8.

This leads to a final score 2, 6, 6, 8, with an average score of 5.5, which is slightly above the borderline threshold.

---

### Decision · Program_Chairs · 2026-01-26

Accept (Poster)